# SeeClear: Semantic Distillation Enhances Pixel Condensation for Video Super-Resolution

**Qi Tang**[1,2], **Yao Zhao**[1,2], **Meiqin Liu**[1,2*], **Chao Yao**[3*]

[1] Institute of Information Science, Beijing Jiaotong University
[2] Visual Intelligence + X International Cooperation Joint Laboratory of MOE,
Beijing Jiaotong University
[3] School of Computer and Communication Engineering,
University of Science and Technology Beijing
{qitang, yzhao, mqliu}@bjtu.edu.cn, yaochao@ustb.edu.cn

## Abstract

Diffusion-based Video Super-Resolution (VSR) is renowned for generating perceptually realistic videos, yet it grapples with maintaining detail consistency across frames due to stochastic fluctuations. The traditional approach of pixel-level alignment is ineffective for diffusion-processed frames because of iterative disruptions. To overcome this, we introduce SeeClear–a novel VSR framework leveraging conditional video generation, orchestrated by instance-centric and channel-wise semantic controls. This framework integrates a Semantic Distiller and a Pixel Condenser, which synergize to extract and upscale semantic details from low-resolution frames. The **In**stance-**C**entric **A**lignment **M**odule (InCAM) utilizes video-clip-wise tokens to dynamically relate pixels within and across frames, enhancing coherency. Additionally, the **Cha**nnel-wise **T**exture Ag**g**regation Mem**ory** (CaTeGory) infuses extrinsic knowledge, capitalizing on long-standing semantic textures. Our method also innovates the blurring diffusion process with the ResShift mechanism, finely balancing between sharpness and diffusion effects. Comprehensive experiments confirm our framework's advantage over state-of-the-art diffusion-based VSR techniques. The code is available: https://github.com/Tang1705/SeeClear-NeurIPS24.

## 1 Introduction

Video super-resolution (VSR) is a challenging low-level vision task that involves improving the resolution and visual quality of the given low-resolution (LR) observations and maintaining the temporal coherence of high-resolution (HR) components. Various deep-learning-based VSR approaches [1, 3, 30, 17, 13, 20, 21] explore effective inter-frame alignment to reconstruct satisfactory sequences. Despite establishing one new pole after another in the quantitative results, they struggle to generate photo-realistic textures.

With the explosion of diffusion model (DM) in visual generation [10, 31, 26], super-resolution (SR) from the generative perspective also garners the broad attention [28, 29, 45, 5]. DM breaks the generation process into sequential sub-processes and iteratively samples semantic-specific images from Gaussian noise, equipped with a paired forward diffusion process and reverse denoising process. The former progressively injects varied intensity noise into the image along a Markov chain to simulate diverse image distributions. The latter leverages a denoising network to generate an image based on the given noise and conditions. Early efforts directly apply the generation paradigm to super-resolution, overlooking its characteristic while generating pleasing content, thus trapping in huge sampling overhead.

---

[*]Corresponding Authors

38th Conference on Neural Information Processing Systems (NeurIPS 2024).

Different from generation from scratch, SR resembles partial generation. The structural information that dominates the early stages of diffusion is contained in the LR priors, while SR tends to focus on generating high-frequency details [33, 15]. Besides, the loss of high-frequency information in LR videos stems from the limited sensing range of imaging equipment. As a result, solely disrupting frames with additive noise is inadequate to depict the degradation of HR videos [9]. Moreover, prevalent VSR methods employ delicate inter-frame alignment (e.g., optical flow or deformable convolution) to fuse the sub-pixel information across adjacent frames. However, the disturbed pixels pose a severe challenge to these methods, rendering the accuracy to deteriorate in the pixel space.

To alleviate the above issue, we introduce **SeeClear**, an innovative diffusion model empowering distilled semantics to enhance the pixel condensation for video super-resolution. During the forward diffusion process, the low-pass filter is applied within the patch, gradually diminishing the high-frequency component, all while the residual is progressively shifted in the frequency domain to transform the HR frames to corresponding LR versions step by step. To reduce the computational overhead, the intermediate states are decomposed into various frequency sub-bands via 2D discrete wavelet transform and subsequently processed by the attention-based U-Net. Furthermore, we devise a dual semantic-controlled conditional generation schema to enhance the temporal coherence of VSR. Specifically, a segmentation framework for open vocabulary is employed to distill instance-centric semantics from LR frames. They serve as prompts,

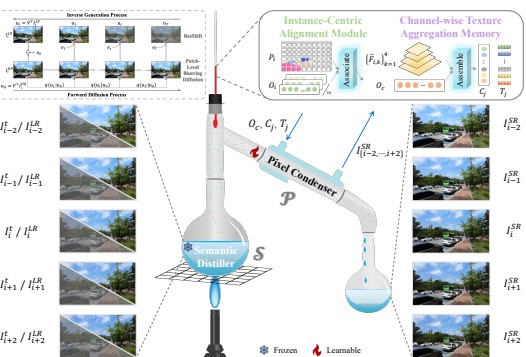

Figure 1: The sketch of SeeClear. It consists of a Semantic Distiller and a Pixel Condenser, which are responsible for distilling instance-centric semantics from LR frames and generating HR frames. The instance-centric and assembled channel-wise semantics act as thermometer to control the condition for generation.

enabling the Instance-Centric Alignment Module (InCAM) to highlight and associate semantically related pixels within the local temporal scope. Besides, abundant semantic cues in channel dimensions are also explored to form an extensional memory dubbed Channel-wise Texture Aggregation Memory (CaTeGory). It aids in global temporal coherence and boosts performance. Experimental results demonstrate that our method consistently outperforms existing state-of-the-art methods.

In summary, the main contributions of this work are as follows:

- We present SeeClear, a diffusion-based framework for video super-resolution that distills semantic priors from low-resolution frames for spatial modulation and temporal association, controlling the condition of pixel generation.

- We reformulate the diffusion process by integrating residual shifting with patch-level blurring, and introduce an attention-based architecture to explore valuable information among wavelet spectra during the sampling process, incorporating feature modulation of intra-frame semantics.

- We devise a dual semantic distillation schema that extracts instance-centric semantics of each frame and further assembles them into texture memory based on the semantic category of channel dimension, ensuring both short-term and long-term temporal coherence.

## 2 Related Work

### 2.1 Video Super-Resolution

Prominent video super-resolution techniques concentrate on leveraging sub-pixel information across frames to enhance performance. EDVR [37] employs cascading deformable convolution layers (DCN) for inter-frame alignment in a coarse-to-fine manner, tackling large amplitude video motion. BasicVSR [1] comprehensively explores each module's role in VSR and delivers a simple yet effective framework by reusing previous designs with slight modifications. Given the similarity between DCN and optical flow, BasicVSR++ [3] devises flow-guided deformable alignment, exploiting the offset diversity of DCN without instability during the training. VRT [16] combines mutual attention with self-attention, which is respectively in charge of inter-frame alignment and information preservation.

RVRT [17] extends this by incorporating optical flow with deformable attention, aligning and fusing features directly at non-integer locations clip-to-clip. PSRT [30] reassesses prevalent alignment methods in transformer-based VSR and implements patch alignment to counteract inaccuracies in motion estimation and compensation. DFVSR [8] represents video with the proposed directional frequency representation, amalgamating object motion into multiple directional frequencies, augmented with a frequency-based implicit alignment, thus enhancing alignment.

## 2.2 Diffusion-Based Super-Resolution

Building on the success of diffusion models in the realm of image generation [10, 31, 26, 25, 48], diffusion-based super-resolution (SR) is advancing. SR3 [28], a pioneering approach, iteratively samples an HR image from Gaussian noise conditioned on the LR image. In contrast, StableSR [36] applies diffusion-based SR in a low-dimensional latent space using the pre-trained auto-encoder to reduce computation and generate improved results through the generative priors contained in weights of Latent Diffusion. ResDiff [29] combines a lightweight CNN with DM to restore low-frequency and predict high-frequency components, and ResShift [45] redefines the initial step as a blend of the low-resolution image and random noise to boost efficiency. Applying a different approach, DiWa [23] migrates the diffusion process into the wavelet spectrum to effectively hallucinate high-frequency information. Upscale-A-Video [49], for video super-resolution, introduces temporal layers into the U-Net and VAE-Decoder and deploys a flow-guided recurrent latent propagation module to ensure temporal coherence and overall video stability when applying image-wise diffusion model.

## 2.3 Semantic-Assisted Restoration

Traditionally seen as a preparatory step for subsequent tasks [50, 12], restoration is now reformulated with the assistance of semantics. SFT [38] utilizes semantic segmentation probability maps for spatial modulation of intermediate features in the SR network, yielding more realistic textures. SKF [40] supports low-light image enhancement model to learn diverse priors encapsulated in a semantic segmentation model by semantic-aware embedding module paired with semantic-guided losses. SeD [14] integrates semantics into the discriminator of GAN-based SR for fine-grained texture generation rather than solely learning coarse-grained distribution. CoSeR [32] bridges image appearance and language understanding to empower SR with global cognition buried in LR image, regarding priors of text-to-image (T2I) diffusion model and a high-resolution reference image as powerful conditions. SeeSR [39] analyzes several types of semantic prompts and opts tag-style semantics to harness the generative potential of the T2I model for real SR. Semantic Lens [34] forgoes pixel-level inter-frame alignment and distills diverse semantics for temporal association in the instance-centric semantic space, attaining better performance.

## 3 Methodology

Given a low-resolution (LR) video sequence of $N$ frames $I_i^{LR} \in \mathbb{R}^{N \times C \times H \times W}$, where $i$ is the frame index, $H \times W$ represents spatial dimensions, and $C$ stands for the channel of frame, SeeClear aims to exploit rich semantic priors to generate the high-resolution (HR) video $I_i^{HR} \in \mathbb{R}^{N \times C \times sH \times sW}$, with $s$ as the upscaling factor. In the iterative paradigm of the diffusion model, HR frames are corrupted according to handcrafted transition distribution at each diffusion step ($t = 1, 2, \cdots, T$). And a U-shaped network is employed to estimate the posterior distribution using LR frames as condition during reverse generation. As illustrated in Figure 1, it consists of a Semantic Distiller and a Pixel Condenser, respectively responsible for semantic extraction and texture generation.

The LR video is initially split into non-overlapping clips composed of $m$ frames for parallel processing. Semantic Distiller, a pre-trained network for open-vocabulary segmentation, distills semantics related to both instances and background clip by clip, denoted as instance-centric semantics. Pixel Condenser is an attention-based encoder-decoder architecture, in which the encoder extracts multi-scale features under the control of LR frames, and the decoder generates HR frames from coarse to fine. They are also bridged via skip connections to transmit high-frequency information at the same resolution. To maximize the network's generative capacity, instance-centric semantics are utilized as conditions for individual frame generation in the decoder. They also serve as the cues of inter-frame alignment for temporal coherence within the video clip and further cluster into a semantic-texture memory along channel dimension for consistency across clips.

### 3.1 Blurring ResShift

During the video capturing, frequencies exceeding the imaging range of the device are truncated, leading to the loss of high-frequency information in LR videos. Therefore, an intuition is to construct a Markov chain between HR frames and LR frames in the frequency domain. Inspired by blurring diffusion [11], the forward diffusion process of SeeClear initializes with the approximate distribution of HR frames. It then iterates and terminates with the approximate distribution of LR frames using a Gaussian kernel convolution in frequency space facilitated by the Discrete Cosine Transformation (DCT). Considering the correlation of neighboring information, blurring is conducted within a local patch instead of the whole image. The above process is formulated as:

$$q\left(\boldsymbol{u}_t \mid \boldsymbol{u}_0\right) = \mathcal{N}\left(\boldsymbol{u}_t \mid \boldsymbol{D}_t \boldsymbol{u}_0, \eta_t \boldsymbol{E}\right), \quad t \in \{1, \cdots, T\}, \tag{1}$$

$$\boldsymbol{u}_0 = \boldsymbol{V}^{\mathrm{T}} I_i^{HR}, \tag{2}$$

where $\boldsymbol{u}_0$ and $\boldsymbol{u}_t$ denote HR frames and intermediate states in the frequency space for brevity. $\boldsymbol{V}^{\mathrm{T}}$ denotes the projection matrix of DCT. $\boldsymbol{D}_t = e^{\boldsymbol{\Lambda} t}$ is diagonal blurring matrix with $\boldsymbol{\Lambda}_{x \times p+y} = -\pi^2\left(\frac{x^2}{p^2} + \frac{y^2}{p^2}\right)$ for coordinate $(x, y)$ within patch of size $p \times p$, and $\eta_t$ is the variance of noise. $\boldsymbol{E}$ is the identity matrix.

In the realm of generation, the vanilla destruction process progressively transforms the image into pure Gaussian noise, leading to numerous sampling steps and tending to be suboptimal for VSR. An alternative way is to employ a transition kernel that shifts residuals between HR and LR frames, accompanied by patch-level blurring. The forward diffusion process is formulated as:

$$q\left(\boldsymbol{u}_t \mid \boldsymbol{u}_0, \boldsymbol{u}_l\right) = \mathcal{N}\left(\boldsymbol{u}_t \mid \boldsymbol{D}_t \boldsymbol{u}_0 + \eta_t \boldsymbol{e}_t, \kappa^2 \eta_t \boldsymbol{E}\right), \quad t \in \{1, \cdots, T\}, \tag{3}$$

$$\boldsymbol{e}_t = \boldsymbol{u}_l - \boldsymbol{D}_t \boldsymbol{u}_0, \tag{4}$$

where $\boldsymbol{u}_l$ denotes LR frames transformed into the frequency space. $\boldsymbol{e}_t$ indicates the residuals between LR and blurred HR frames at time step $t$. $\eta_t$ represents the shifting sequence and $\kappa$ is a hyper-parameter determining the intensity of noise. Upon this, SeeClear can yield HR frames by estimating the posterior distribution $p(\boldsymbol{u}_0|\boldsymbol{u}_l)$ in the reverse sampling progress, formulated as:

$$p\left(\boldsymbol{u}_0 \mid \boldsymbol{u}_l\right) = \int p\left(\boldsymbol{u}_T \mid \boldsymbol{u}_l\right) \prod_{t=1}^{T} p_{\boldsymbol{\theta}}\left(\boldsymbol{u}_{t-1} \mid \boldsymbol{u}_t, \boldsymbol{u}_l\right) \mathrm{d}\boldsymbol{u}_{1:T}, \tag{5}$$

$$p\left(\boldsymbol{u}_T \mid \boldsymbol{u}_l\right) \approx \mathcal{N}\left(\boldsymbol{u}_T \mid \boldsymbol{u}_l, \kappa^2 \boldsymbol{E}\right), \tag{6}$$

where $p_{\boldsymbol{\theta}}\left(\boldsymbol{u}_{t-1} \mid \boldsymbol{u}_t, \boldsymbol{u}_l\right)$ represents the inverse transition kernel restoring $\boldsymbol{u}_t$ to $\boldsymbol{u}_{t-1}$. $\theta$ denotes learnable parameters of attention-based U-Net.

To alleviate the computational overhead, preceding methods introduce an autoencoder to transform pixel-level images in the perceptually equivalent space, concentrating on the semantic composition and bypassing the impedance of high-frequency details. However, the loss of high-frequency information during encoding is hard to recover in the decoding and will deteriorate the visual quality. Therefore, we forgo the autoencoding method in SeeClear and incorporate discrete wavelet transform (DWT) in the diffusion process. Specifically, the HR and LR frames are recursively decomposed into four sub-bands:

$$I_{ll}^{HR}, I_{lh}^{HR}, I_{hl}^{HR}, I_{hh}^{HR} = \mathrm{DWT}_{2\mathrm{D}}\left(I_i^{HR}\right), \tag{7}$$

where $I_{ll}^{HR}$ denotes the low-frequency approximation, $I_{lh}^{HR}$, $I_{hl}^{HR}$ and $I_{hh}^{HR}$ correspond to horizontal, vertical and diagonal high-frequency details. $\mathrm{DWT}_{2\mathrm{D}}(\cdot)$ represents the 2D Discrete Wavelet Transform (DWT). After $k$ decompositions, each of them possesses a size of $\frac{H}{2^k} \times \frac{W}{2^k}$. These coefficients are contacted along the channel dimension and serve in the diffusion process. The rationale behind employing DWT as a substitute is two-fold. Firstly, it enables the U-Net to perform on a small spatial size without information loss. Secondly, it benefits from the U-Net scaling [44] hindered by additional parameters of the autoencoding network. To make full use of DWT, window-based self-attention followed by channel-wise self-attention is stacked as the basic unit of U-Net, in charge of the correlation of intra-sub-bands and inter-sub-bands, respectively.

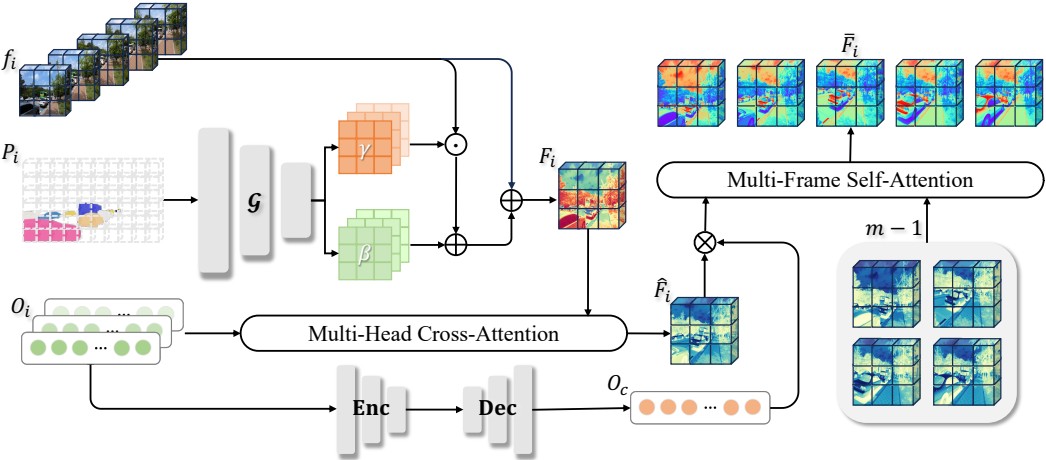

Figure 2: The illustration of Instance-Centric Alignment Module (InCAM). It utilizes the segmentation features to bridge the pixel-level information and instance-centric semantic tokens. And then, the semantic-aware features can be aligned in the semantic space based on their semantic relevance.

## 3.2 Instance-Centric Alignment within Video Clips

Due to the destruction of the diffusion process, pixel-level inter-frame alignment, such as optical flow, is no longer applicable. With the premise of semantic embedding, we devise the Instance-Centric Alignment Module (InCAM) within video clips, as illustrated in Figure 2. It establishes temporal association in the semantic space instead of intensity similarity among frames, avoiding the interference of noise and blurriness. Specifically, Semantic Distiller predicts a set of image features $F_{img}$ and text embedding $F_{txt}$ from LR frames and predefined vocabulary $\mathcal{V}$. After that, the $k$ image features with the highest similarity to the text embedding are retained, including token-level semantic priors and pixel-level segmentation features from LR frames. The above procedure is formulated as:

$$F_{img}, F_{txt} = \mathcal{S}\left(I_i^{LR}, \mathcal{V}\right), \tag{8}$$

$$O_i, P_i = top_k\left(\text{Sim}\left(F_{img}, F_{txt}\right)\right), \tag{9}$$

where $\mathcal{S}\left(\cdot\right)$ represents Semantic Distiller. $top_k\left(\cdot\right)$ and $\text{Sim}\left(\cdot\right)$ denote the operations of selecting the $k$ largest items and calculating the similarity respectively. $O_i$ and $P_i$ are semantic tokens and segmentation features, in which the former represents high-level semantics and can locate related pixels in the segmentation features. The segmentation features contain both semantics and low-level structural information, which is suitable for bridging the semantics and features of the Pixel Condenser. It is utilized to generate spatial modulation pairs prepared for semantic embedding, formulated as:

$$(\gamma, \beta) = \mathcal{G}\left(P_i\right), \tag{10}$$

$$F_i = (f_i \odot \gamma + \beta) + f_i, \tag{11}$$

where $\gamma$ and $\beta$ represent scale and bias for modulation. $f_i$ and $F_i$ correspond to original and modulated features. $\mathcal{G}$ denotes two convolutional layers followed by a ReLU activation. "$\odot$" represents the Hadamard product. After that, InCAM embeds semantics into modulated features based on multi-head cross-attention, yielding semantic-embedded features $\hat{F}_i$:

$$\mathbf{Q}_i = F_i W^Q, \ \mathbf{K}_i = O_i W^K, \ \mathbf{V}_i = O_i W^V, \tag{12}$$

$$\hat{F}_i = \text{SoftMax}\left(\mathbf{Q}_i \mathbf{K}_i^T / \sqrt{d}\right) \mathbf{V}_i, \tag{13}$$

where $\mathbf{Q}_i$, $\mathbf{K}_i$ and $\mathbf{V}_i$ denote matrices derived from modulated features and semantic tokens. $W^Q$, $W^K$ and $W^V$ represent the linear projections, and $d$ is the dimension of projected matrices. $\text{SoftMax}\left(\cdot\right)$ denotes the SoftMax operation. To benefit from the adjacent supporting frames, it is necessary to establish semantic associations among frames. The frame-wise semantic tokens are further fed into the instance encoder-decoder for communicating semantics along the temporal

dimension, generating clip-wise semantic tokens. It gathers all the information of the same semantic object within a clip and serves as the guide for inter-frame alignment. Specifically, InCAM combines semantic guidance and enhanced features to activate the related pixels across frames and utilizes multi-frame self-attention for parallel alignment and fusion. The above procedure is formulated as:

$$O_c = \text{Dec}\left(\text{Enc}\left(O_i\right), \hat{O}\right), \tag{14}$$

$$\bar{F}_i = \text{MFSA}\left(O_c \cdot \hat{F}_i\right), \tag{15}$$

where $O_c$ and $\hat{O}$ respectively denote clip-wise semantic and randomly initialized tokens. $\text{Enc}\left(\cdot\right)$ and $\text{Dec}\left(\cdot\right)$ represent instance encoder and decoder. $\text{MFSA}\left(\cdot\right)$ denotes the multi-frame self-attention [30], the extended version of self-attention in video. $\bar{F}_i$ is aligned feature. The product of semantics and enhanced features is akin to the class activation mapping, which highlights the most similar pixels in the instance-centric semantic space among frames.

### 3.3 Channel-wise Aggregation across Video Clips

Due to the limited size of the temporal window, the performance of clip-wise mutual information enhancement in long video sequences is unsatisfactory, which could lead to inconsistent content. To stabilize video content and enhance visual texture, the Channel-wise Texture Aggregation Memory (CaTeGory) is constructed to cluster abundant textures according to channel-wise semantics, as graphically depicted in Figure 3. It is comprised of the channel-wise semantic and the corresponding texture. Specifically, channels of instance-centric semantics also contain distinguishing traits, which are assembled and divided into different groups to form the channel-wise semantic. Concurrently, hierarchical features from the decoder of the Pixel Condenser are clustered into the corresponding semantic group to portray the textures. The connection between them is established in a manner similar to the position embedding in the attention mechanism. The above process can be formulated as:

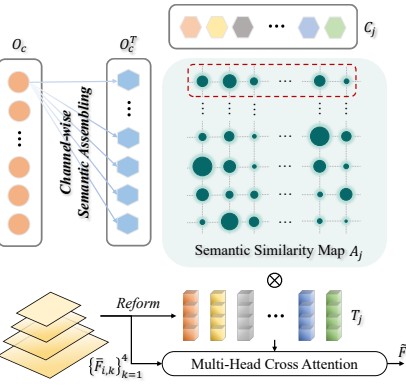

Figure 3: The illustration of Channel-wise Texture Aggregation Memory (CaTeGory). It assembles the textures based on the semantic class along the channel dimension.

$$\left(C_j, T_j\right) = \mathcal{M}\left(\bar{C}_j, \bar{T}_j, \left\{\bar{F}_{i,k}\right\}_{k=1}^{4}\right), \tag{16}$$

where $\bar{F}_{i,k}$ is the features benefited from adjacent frames of $k$-th layer. $\bar{C}_j$ and $\bar{T}_j$ respectively denote channel-wise semantics and textures of $j$-th group, which are zero-initialized as network parameters. They are iteratively updated towards the final version (i.e., $C_j$ and $T_j$) by injecting external knowledge from the whole dataset and previous clips. And $\mathcal{M}\left(\cdot\right)$ represents the construction of CaTeGory. It concatenates the multi-scale features and incorporates them into channel-wise semantics and textures:

$$\hat{T}_j = \bar{C}_j \times \bar{T}_j, \tag{17}$$

$$T_j = \text{SA}\left(\text{CA}\left(\hat{T}_j, \left\{\bar{F}_{i,k}\right\}_{k=1}^{4}\right)\right), \tag{18}$$

where $\hat{T}_j$ is the textures embedded channel-wise semantics. $\text{SA}\left(\cdot\right)$ and $\text{CA}\left(\cdot\right)$ indicate multi-head self-attention and cross-attention. The layer normalization and feed-forward network are omitted for brevity. It bridges the channel-wise semantics and textures via matrix multiplication and further fuses high-value information from the pyramid feature, delivering augmented semantic-texture pairs. The hierarchical features not only provide rich structural information but also carry relatively abstract amid features in order to benefit different decoder layers more effectively. At each layer, the prior knowledge stored in CaTeGory is firstly queried by the clip-wise semantics along channel dimension and aggregated for feature enhancement of the current clip, which is formulated as:

$$\mathbf{A}_j = \text{SoftMax}\left(O_c^T C_j\right), \tag{19}$$

$$\tilde{F}_i = \text{CA}\left(\bar{F}_i, \mathbf{A}_j T_j\right) + \bar{F}_i, \tag{20}$$

where $\mathbf{A}_j$ depicts the similarity between the clip-wise semantics and items of CaTeGory along channel dimensions, and $\tilde{F}_i$ is the refined features as input of the next layer. As mentioned before, semantic-texture pairs are optimized as parts of the network during the training stage, absorbing ample priors from the whole dataset. In the sampling process, the update mechanism is reused to integrate the long-term information of video into memory to improve the super-resolution of subsequent clips.

# 4 Experiments

## 4.1 Experimental Setup

**Datasets** To assess the effectiveness of the proposed SeeClear, we employ two commonly used datasets for training: REDS [24] and Vimeo-90K [41]. The REDS dataset, characterized by its realistic and dynamic scenes, consists of three subsets used for training and testing. In accordance with the conventions established in previous works [1, 3], we select four clips[2] from the training dataset to serve as a validation dataset, referred to as REDS4. The Vid4 [19] dataset is used as the corresponding test dataset for Vimeo-90K. The LR sequences are degraded through bicubic downsampling (BI), with a downsampling factor of $4\times$.

**Implementation Details** The pre-trained OpenSeeD [46] is opted as the semantic distiller with frozen weights, while all learnable parameters are contained in the pixel condenser. And the schedulers of blur and noise in the diffusion process follow the settings of IHDM [25] and ResShift [45]. During the training, the pixel condenser is first trained to generate an HR clip with 5 frames under the control of instance-centric semantics. And then, Channel-wise Texture Memory is independently trained to inject valuable textures from the whole dataset and be capable of fusing long-term information. Finally, the whole network is jointly fine-tuned. All training stages utilize the Adam optimizer with $\beta_1 = 0.5$ and $\beta_2 = 0.999$, where the learning rate decays with the cosine annealing scheme. The Charbonnier loss [4] is applied on the whole frames between the ground truth and the reconstructed frame, formulated as $L = \sqrt{||I_i^{HR} - I_i^{SR}||^2 + \epsilon^2}$. The SeeClear framework is implemented with PyTorch-2.0 and trained across 4 NVIDIA 4090 GPUs, each accommodating 4 video clips.

**Evaluation Metrics** Comparative analysis is conducted among different VSR methods, with the evaluation being anchored on both pixel-based and perception-oriented metrics. Peak Signal-to-Noise Ratio (PSNR) and Structural Similarity Index (SSIM) are utilized to evaluate the quantitative performance as pixel-based metrics. All of them are calculated based on the Y-channel, with the exception of the REDS4, for which the RGB-channel is used. On the perceptual side, Learned Perceptual Image Patch Similarity (LPIPS) [47] is elected for assessment from the perspective of human visual preference. It leverages a VGG model to extract features from the generated HR video and the ground truth, subsequently measuring the extent of similarity between these features.

## 4.2 Comparisons with State-of-the-Art Methods

We compare SeeClear with several state-of-the-art methods, including regression-based and diffusion-based ones. As shown in Table 3, SeeClear achieves superior perceptual quality compared to regression-based methods despite slightly underperforming in pixel-based metrics. We also provide an extended version, which leverages the generative capability of SeeClear to enhance the features of the regression-based model, akin to references such as [6, 2]. An observable increase in fidelity is accompanied by a notable further improvement in the perceptual metrics of the reconstructed results. Similar performance trends can be noted on Vid4 as those on REDS4. In particular, SeeClear achieves an LPIPS score of 0.1548, marking a relative improvement of 10.8% compared to the top competitor, SATeCo [6]. When pitted against a variant of SATeCo, which is not modulated by LR videos, SeeClear demonstrates a higher PSNR value with a comparable LPIPS score. It suggests that SeeClear benefits from the control of dual semantics, striking a balance between superior fidelity and the generation of realistic textures.

As visualized in Figure 4, SeeClear showcases its ability to restore textures with high fidelity more effectively compared with other methods. Despite large blurriness, SeeClear still demonstrates robust restoration capabilities for video super-resolution, reinforcing the efficacy of utilizing instance-specific and channel-wise semantic priors for video generation control. To further substantiate

---

[2]Clip 000, 011, 015, 020

Table 1: Performance comparisons in terms of pixel-based (PSNR and SSIM) and perception-oriented (LPIPS) evaluation metrics on the REDS4 [24] and Vid4 [19] datasets. The extended version of SeeClear is marked with ⋆. Red indicates the best, and blue indicates the runner-up performance (best view in color) in each group of experiments.

| Methods | Frames | REDS4 [24] | | | Vid4 [19] | | |
|---|---|---|---|---|---|---|---|
| | | PSNR ↑ | SSIM ↑ | LPIPS ↓ | PSNR ↑ | SSIM ↑ | LPIPS ↓ |
| Bicubic | - | 26.14 | 0.7292 | 0.3519 | 23.78 | 0.6347 | 0.3947 |
| TOFlow [41] | 7 | 29.98 | 0.7990 | 0.3104 | 25.89 | 0.7651 | 0.3386 |
| EDVR-M [37] | 5 | 30.53 | 0.8699 | 0.2312 | 27.10 | 0.8186 | 0.2898 |
| BasicVSR [1] | 15 | 31.42 | 0.8909 | 0.2023 | 27.24 | 0.8251 | 0.2811 |
| VRT [16] | 6 | 31.60 | 0.8888 | 0.2077 | 27.93 | 0.8425 | 0.2723 |
| IconVSR [1] | 15 | 31.67 | 0.8948 | 0.1939 | 27.39 | 0.8279 | 0.2739 |
| StableSR [36] | 1 | 24.79 | 0.6897 | 0.2412 | 22.18 | 0.5904 | 0.3670 |
| ResShift [45] | 1 | 27.76 | 0.8013 | 0.2346 | 24.75 | 0.7040 | 0.3166 |
| SATeCo [6] | 6 | 31.62 | 0.8932 | 0.1735 | 27.44 | 0.8420 | 0.2291 |
| SeeClear (Ours) | 5 | 28.92 | 0.8279 | 0.1843 | 25.63 | 0.7605 | 0.2573 |
| SeeClear⋆ (Ours) | 5 | 31.32 | 0.8856 | 0.1548 | 27.80 | 0.8404 | 0.2054 |

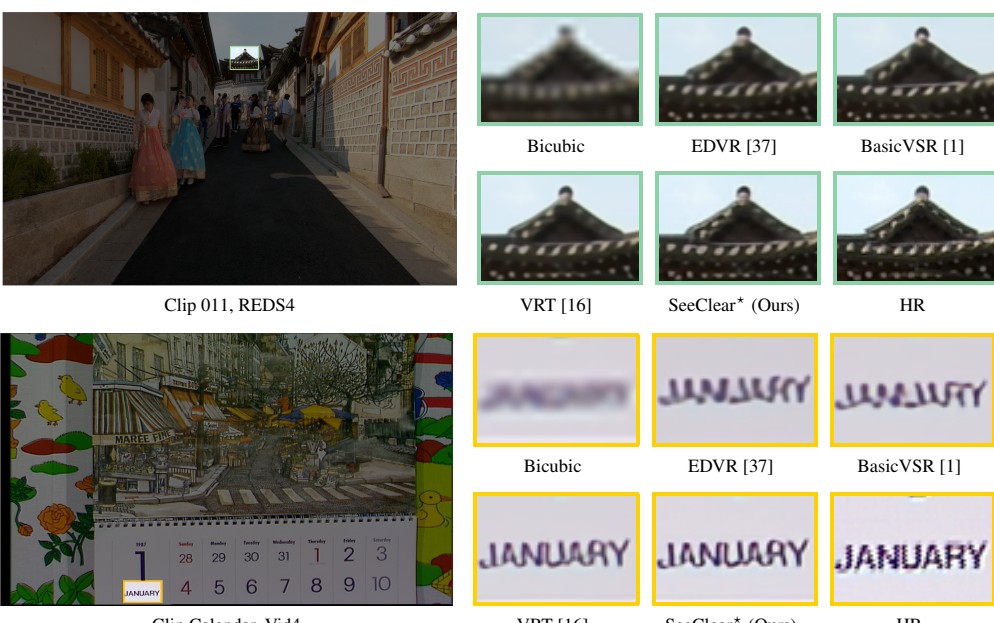

Figure 4: Qualitative results on the REDS4 and Vid4 datasets. SeeClear generates clearer content and sharper textures.

the temporal coherence acquired by SeeClear, we also visualize two consecutive frames from the generated HR videos, constructed using different diffusion-based VSR methodologies, as depicted in Figure 5. ResShift synthesizes varied visual contents across two frames, such as the fluctuating figures on the license plate. Contrarily, HR frames generated via SeeClear maintain a higher temporal consistency and deliver pleasing textures.

## 4.3 Ablation Study

To assess the contribution of each component within the proposed SeeClear, we begin with a baseline model and gradually integrate these modules. Specifically, all semantic-related operations are bypassed, retaining solely spatial and channel self-attention and residual blocks, degenerating into a diffusion-based image SR model without any condition. Subsequently, we incrementally introduce the crafted semantic-conditional module into the baseline and formulate several variants. Their results are listed in Table 2 and partially visualized in Figure 6.

Table 2: Performance comparisons on REDS4 among variants with different semantic-condition control by integrating InCAM and CaTeGory.

| | Baseline | DWT | Semantic | MFSA | InCAM | CaTeGory | PSNR ↑ | SSIM ↑ | LPIPS ↓ |
|---|---|---|---|---|---|---|---|---|---|
| 1 | ✓ | ✓ | | | | | 28.05 | 0.7993 | 0.2120 |
| 2 | ✓ | ✓ | ✓ | | | | 28.08 | 0.7998 | 0.2088 |
| 3 | ✓ | ✓ | ✓ | ✓ | | | 27.99 | 0.7961 | 0.2053 |
| 4 | ✓ | ✓ | ✓ | ✓ | ✓ | | 28.46 | 0.8098 | 0.1917 |
| 5 | ✓ | ✓ | ✓ | | | ✓ | 28.21 | 0.7986 | 0.2149 |
| 6 | ✓ | | ✓ | ✓ | ✓ | ✓ | 28.74 | 0.8267 | 0.1938 |
| 7 | ✓ | ✓ | ✓ | ✓ | ✓ | ✓ | 28.92 | 0.8279 | 0.1843 |

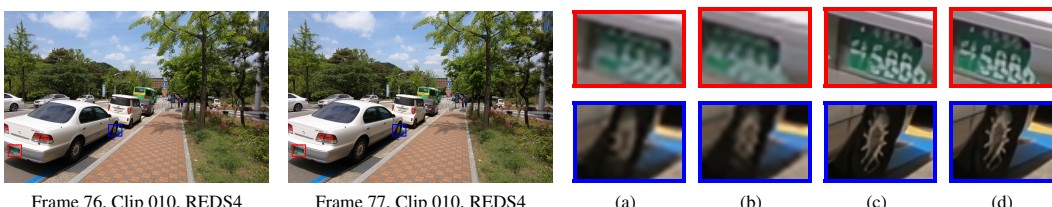

Frame 76, Clip 010, REDS4    Frame 77, Clip 010, REDS4    (a)    (b)    (c)    (d)

Figure 5: Qualitative comparison of regions between consecutive frames. (a) and (b) are patches produced by ResShift [45], derived from Frames 76 and 77 respectively. (c) and (d) display the corresponding regions as generated through SeeClear.

First, the intra-frame semantic condition brings about 1.5% improvements in LPIPS. Albeit the multi-frame self-attention further improves the perceptual quality, it also impairs the fidelity of the restored video. Under the control of InCAM, SeeClear can correlate semantically consistent pixels in adjacent frames by combining intra-frame and inter-frame semantic priors, elevating the PSNR from 27.99 dB to 28.46 dB, and bringing about 6.6% improvements in LPIPS. Furthermore, upon integrating the semantic priors from CaTeGory, the fully-fledged SeeClear notably enhances both the pixel-based and perception-oriented metrics simultaneously. It indicates that the cooperative control of semantics is more

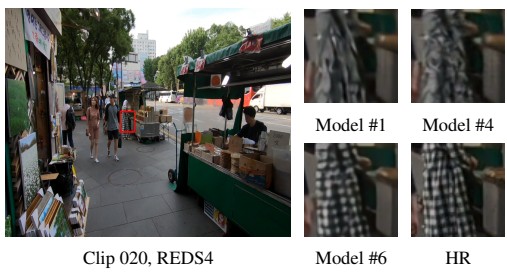

Model #1    Model #4

Clip 020, REDS4    Model #6    HR

Figure 6: Visual comparisons of ablation for investigating the contribution of key modules.

beneficial for generating videos of higher fidelity and better perceptual quality. As illustrated in Figure 6, the baseline struggles to restore tiny and fine patterns without semantic condition, and it gradually gains improvement accompanied by the strengthening of semantic control.

## 5    Conclusion

In this work, we present a novel diffusion-based video super-resolution framework named SeeClear. It formulates the diffusion process by incorporating residual shifting mechanism and patch-level blurring, constructing a Markov chain initiated with high-resolution frames and terminated at low-resolution frames. It employs a semantic distiller and a pixel condenser for super-resolution during the inverse sampling process. The instance-centric semantics distilled by the semantic distiller prompts spatial modulation and temporal association in the devised Instance-Centric Alignment Module. They are further assembled into Channel-wise Texture Aggregation Memory, providing abundant conditions for temporal coherence and realistic content.

**Acknowledgment.** This work is supported in part by the National Natural Science Foundation of China under Grant 62120106009 and Grant 62372036; and in part by the National Key Research and Development Program of China under Grant 2022ZD0118001 and Grant 2021ZD0112100.

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

## A  Mathematical Details

**Patch-level Blurring Diffusion.** Blurring Diffusion [11] designs the destruction process based on heat dissipation [25] that retains low-frequency components of images over high frequencies. It describes the thermodynamic process in which the temperature $u(x, y, t)$ changes at position $(x, y)$ in a 2D plane with respect to the time $t$ via a partial differential equation:

$$\frac{\partial}{\partial t} \boldsymbol{u}(x, y, t) = \Delta \boldsymbol{u}(x, y, t), \quad t \in \{1, \cdots, T\}, \tag{21}$$

where $\Delta = \nabla^2$ denotes the Laplace operator. Based on Neumann boundary conditions ($\partial u/\partial x = \partial u/\partial y = 0$) with zero-derivatives at the boundaries of the image, the solution is given by a diagonal matrix in the frequency domain of the discrete cosine transform:

$$\boldsymbol{u}_t = e^{\boldsymbol{\Lambda} t} \boldsymbol{u}_0, \quad t \in \{1, \cdots, T\}, \tag{22}$$

where $\boldsymbol{u}_0$ is the initial state and $\boldsymbol{\Lambda}$ is a diagonal matrix with negative squared frequencies on the diagonal. It is equivalent to a convolution with a Gaussian kernel with variance $\sigma_{blur}^2 = 2t$ in the realm of image processing. Blurring Diffusion further mixes Gaussian noise with variance $\sigma_t^2$ into the blurring process to incorporate stochasticity into deterministic dissipation.

Patch-level blurring diffusion conducts the blurring process within a patch with the size of $p \times p$ to make all pixel intensity the same instead of the whole image, which means $\boldsymbol{\Lambda}_{x \times p+y} = -\pi^2(\frac{x^2}{p^2} + \frac{y^2}{p^2})$. More generally, it can be extended with any invertible transformation, corresponding to the marginal distribution in the pixel space:

$$q\left(I_i^t \mid I_i^{HR}\right) = \mathcal{N}\left(I_i^t \mid \boldsymbol{R} \operatorname{diag}(\alpha_t) \boldsymbol{R}^{-1} I_i^{HR}, \boldsymbol{R} \operatorname{diag}(\beta_t) \boldsymbol{R}^{-1}\right), \quad t \in \{1, \cdots, T\}, \tag{23}$$

where $\boldsymbol{R}$ and $\boldsymbol{R}^{-1}$ represent invertible transformation and corresponding inverse transformation. $\operatorname{diag}(\cdot)$ denotes the operation that projects a vector to a diagonal matrix. $\alpha_t$ and $\beta_t$ are specific schedules for mean and noise.

**Blurring ResShift.** ResShift [45] introduces a monotonically increasing sequence $\{\eta_t\}_{t=1}^T$ to gradually shift residual between low-resolution image and high-resolution one, whose marginal distribution is defined as:

$$q\left(I_i^t \mid I_i^0, I_i^{LR}\right) = \mathcal{N}\left(I_i^t \mid I_i^0 + \eta_t e_0, \kappa^2 \eta_t \boldsymbol{E}\right), \quad t \in \{1, \cdots, T\}, \tag{24}$$

where $e_0$ is the residuals between low-resolution and high-resolution frames. $\eta_t$ controls the speed of residual shifting and satisfies $\eta_1 \to 0$ and $\eta_T \to 1$. After incorporation of patch-level blurring diffusion and ResShift, $\boldsymbol{u}_t$ can be sampled via

$$\boldsymbol{u}_t = e^{\boldsymbol{\Lambda} t} \boldsymbol{u}_0 + \eta_t \boldsymbol{e}_t + \kappa \sqrt{\eta_t} \epsilon_t \Leftrightarrow \boldsymbol{u}_t = (1 - \eta_t) e^{\boldsymbol{\Lambda} t} \boldsymbol{u}_0 + \eta_t \boldsymbol{u}_l + \kappa \sqrt{\eta_t} \epsilon_t, \tag{25}$$

$$\boldsymbol{e}_t = \boldsymbol{u}_l - \boldsymbol{D}_t \boldsymbol{u}_0 \Leftrightarrow \boldsymbol{e}_t = \boldsymbol{u}_l - e^{\boldsymbol{\Lambda} t} \boldsymbol{u}_0, \tag{26}$$

where $\epsilon_t \sim \mathcal{N}(\boldsymbol{u} \mid 0, \boldsymbol{E})$. Thus, the relation between $\boldsymbol{u}_t$ and $\boldsymbol{u}_{t-1}$ can be obtained:

$$\hat{\boldsymbol{u}}_0 = \frac{\boldsymbol{u}_{t-1} - \eta_{t-1} \boldsymbol{u}_l}{(1 - \eta_{t-1}) e^{\boldsymbol{\Lambda}(t-1)}}, \tag{27}$$

$$\boldsymbol{u}_t = \frac{1 - \eta_t}{1 - \eta_{t-1}} e^{\boldsymbol{\Lambda}} (\boldsymbol{u}_{t-1} - \eta_{t-1} \boldsymbol{u}_l) + \eta_t \boldsymbol{u}_l + \kappa \sqrt{\alpha_t} \epsilon_t, \tag{28}$$

where $\alpha_t = \eta_t - \eta_{t-1}$. $\hat{\boldsymbol{u}}_0$ is approximate HR frame. By recursively applying the sampling procedure and reparameterization trick, we can rewrite the marginal distribution of Blurring ResShift as follows:

$$q\left(\boldsymbol{u}_t \mid \boldsymbol{u}_0, \boldsymbol{u}_l\right) = \mathcal{N}\left(\boldsymbol{u}_t \mid \boldsymbol{D}_t \boldsymbol{u}_0 + \eta_t \boldsymbol{e}_t, \kappa^2 \eta_t \boldsymbol{E}\right), \quad t \in \{1, \cdots, T\}, \tag{29}$$

According to Bayes's theorem, there is

$$q\left(\boldsymbol{u}_{t-1} \mid \boldsymbol{u}_t, \boldsymbol{u}_0, \boldsymbol{u}_l\right) \propto q\left(\boldsymbol{u}_t \mid \boldsymbol{u}_{t-1}, \boldsymbol{u}_l\right) q\left(\boldsymbol{u}_{t-1} \mid \boldsymbol{u}_0, \boldsymbol{u}_l\right), \tag{30}$$

where

$$q\left(\boldsymbol{u}_t \mid \boldsymbol{u}_{t-1}, \boldsymbol{u}_l\right) = \mathcal{N}\left(\boldsymbol{u}_t \mid \frac{1 - \eta_t}{1 - \eta_{t-1}} e^{\boldsymbol{\Lambda}} (\boldsymbol{u}_{t-1} - \eta_{t-1} \boldsymbol{u}_l) + \eta_t \boldsymbol{u}_l, \kappa^2 \alpha_t \boldsymbol{E}\right), \tag{31}$$

$$q\left(\boldsymbol{u}_{t-1} \mid \boldsymbol{u}_0, \boldsymbol{u}_l\right) = \mathcal{N}\left(\boldsymbol{u}_{t-1} \mid (1 - \eta_{t-1}) e^{\boldsymbol{\Lambda}(t-1)} \boldsymbol{u}_0 + \eta_{t-1} \boldsymbol{u}_l, \kappa^2 \eta_{t-1} \boldsymbol{E}\right), \tag{32}$$

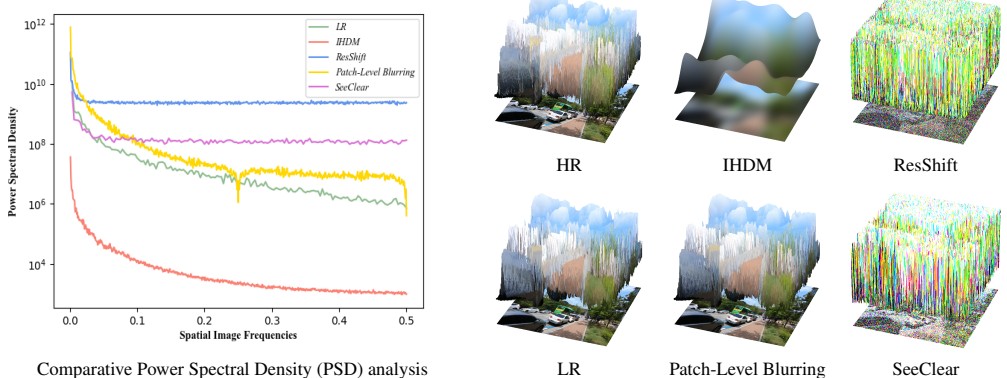

Figure 4: Visual comparison of intermediate state at time step $t$ via different diffusion processes.

And then, it only needs to focus on the quadratic term in the exponent of $q\left(\boldsymbol{u}_{t-1} \mid \boldsymbol{u}_t, \boldsymbol{u}_0, \boldsymbol{u}_l\right)$:

$$
-\frac{\left(\boldsymbol{u}_t - \frac{1-\eta_t}{1-\eta_{t-1}} e^{\boldsymbol{\Lambda}}\left(\boldsymbol{u}_{t-1} - \eta_{t-1}\boldsymbol{u}_l\right) - \eta_t \boldsymbol{u}_l\right)\left(\boldsymbol{u}_t - \frac{1-\eta_t}{1-\eta_{t-1}} e^{\boldsymbol{\Lambda}}\left(\boldsymbol{u}_{t-1} - \eta_{t-1}\boldsymbol{u}_l\right) - \eta_t \boldsymbol{u}_l\right)^T}{2\kappa^2 \alpha_t}
$$

$$
-\frac{\left(\boldsymbol{u}_{t-1} - \left(1-\eta_{t-1}\right) e^{\boldsymbol{\Lambda}(t-1)}\boldsymbol{u}_0 - \eta_{t-1}\boldsymbol{u}_l\right)\left(\boldsymbol{u}_{t-1} - \left(1-\eta_{t-1}\right) e^{\boldsymbol{\Lambda}(t-1)}\boldsymbol{u}_0 - \eta_{t-1}\boldsymbol{u}_l\right)^T}{2\kappa^2 \eta_{t-1}}
$$

$$
= -\frac{1}{2}\left[\frac{\frac{(1-\eta_t)^2}{(1-\eta_{t-1})^2} e^{2\boldsymbol{\Lambda}}}{\kappa^2 \alpha_t} + \frac{1}{\kappa^2 \eta_{t-1}}\right]\boldsymbol{u}_{t-1}\boldsymbol{u}_{t-1}^T + \left[\frac{1-\eta_t}{1-\eta_{t-1}} e^{\boldsymbol{\Lambda}} \frac{\boldsymbol{u}_t + \left(\eta_{t-1}\frac{1-\eta_t}{1-\eta_{t-1}} e^{\boldsymbol{\Lambda}} - \eta_t\right)\boldsymbol{u}_l}{\kappa^2 \alpha_t}\right.
$$

$$
\left.+\frac{\left(1-\eta_{t-1}\right) e^{\boldsymbol{\Lambda}(t-1)}\boldsymbol{u}_0 - \eta_{t-1}\boldsymbol{u}_l}{\kappa^2 \eta_{t-1}}\right]\boldsymbol{u}_{t-1}^T + \text{const}
$$

$$
= -\frac{\left(\boldsymbol{u}_{t-1} - \boldsymbol{\mu}\right)\left(\boldsymbol{u}_{t-1} - \boldsymbol{\mu}\right)^T}{2\sigma^2} + \text{const},
$$

(33)

where

$$
\mu = \frac{\lambda\eta_{t-1}\boldsymbol{u}_t + \alpha_t\left(1-\eta_{t-1}\right) e^{\boldsymbol{\Lambda}(t-1)}\boldsymbol{u}_0 + \left(\lambda^2\eta_{t-1}^2 - \lambda\eta_{t-1}\eta_t - \alpha_t\eta_{t-1}\right)\boldsymbol{u}_l}{\lambda^2\eta_{t-1} + \alpha_t}
$$

(34)

$$
\sigma^2 = \frac{\kappa^2\alpha_t\eta_{t-1}}{\lambda^2\eta_{t-1} + \alpha_t},
$$

(35)

$$
\lambda = \frac{1-\eta_t}{1-\eta_{t-1}} e^{\boldsymbol{\Lambda}},
$$

(36)

where 'const' denotes the item that is independent of $\boldsymbol{u}_{t-1}$.

## B Rational Explanation

We analyze the final states of different diffusion processes via the power spectral density, which reflects the distribution of frequency content in an image, as illustrated in Figure 4. It can be observed that IHDM performs blurring globally and has a significant difference in frequency distribution compared to the LR image, while the patch-level blurring is closer to the frequency distribution of the LR. On this basis, SeeClear further introduces residual and noise. Compared to ResShift without blurring, the diffusion process adopted by SeeClear makes the image more consistent with the LR in the low-frequency components and introduces more randomness in the high-frequency components, compelling the model to focus on the generation of high-frequency components.

## C    Network Structure

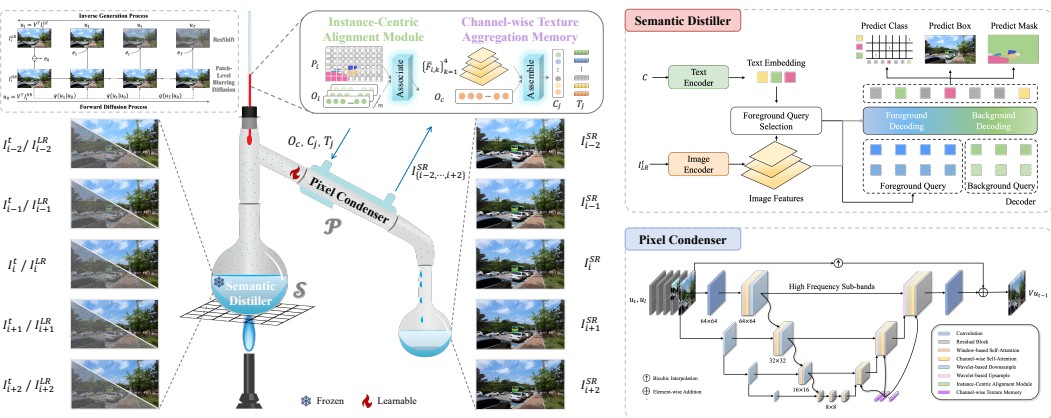

Figure 8: The illustration of SeeClear. It comprises the diffusion process incorporating patch-level blurring and residual shift mechanism and a reverse process. During the reverse process, Semantic Distiller for semantic embedding extraction and U-shaped Pixel Condenser are employed for iterative denoising. The devised InCAM and CaTeGory are inserted into the U-Net to utilize the diverse semantics for inter-frame alignment in the diffusion-based VSR framework.

SeeClear consists of a forward diffusion process and a reverse process for VSR. In the diffusion process, patch-level blurring and residual shift mechanism are integrated to degrade HR frames based on the handcrafted time schedule. During the reverse process, a transformer-based network for open vocabulary segmentation and a U-Net are employed for iterative denoising. The former is responsible for extracting semantic embeddings related to instances from LR videos, similar to the process of distillation in physics, and is therefore named the semantic distiller. The latter is utilized to filter out interfering noise and retain valuable information from low-quality frames, similar to the condensation process. All of them are tailored for image processing, and SeeClear takes diverse semantic embeddings as conditions to enable the network to be aware of the generated content and determine the aligned pixels from adjacent frames for the temporal consistency of the whole video.

As depicted in Figure 8, the attention-based Pixel Condenser primarily consists of three parts, i.e., encoder, decoder, and middle block. As the input of the encoder, low-resolution frames are concatenated with the intermediate states and processed through a convolution layer. The encoder incorporates a window-based self-attention and channel-wise self-attention, alternating between two residual blocks. These attention operations mine valuable information within and across the wavelet sub-bands, and the residual blocks infuse the features with the intensity of degradation. Following this, a wavelet-based downsample layer is deployed for feature downsampling.

Specifically, the features are decomposed into various sub-bands via a 2D discrete wavelet transform, reducing the spatial dimensions while keeping the original data intact. Low-frequency features are further fed into the subsequent layer of the encoder, while others are transmitted to the corresponding wavelet-based upsample layer in the decoder via a skip connection. Additionally, the wavelet-based downsample layers are utilized parallel along the encoder, refilling the downsampled features with information derived from the low-resolution frames.

The devised Instance-Centric Alignment Module (InCAM) and Channel-wise Texture Aggregation Memory (CaTeGory) are inserted into both the middle and decoder. Firstly, the InCAM spatially modulates the features based on instance-centric semantics and aligns adjacent frames within the clip in semantic space. Spatial self-attention employs these aligned features, substituting the original features as input. Subsequently, features enhanced through channel-wise self-attention are embedded as queries to seek assistance from the CaTeGory. Furthermore, the wavelet-based upsample layer accepts features from the decoder and high-frequency information transmitted from the encoder. It reinfuses the lost information from the encoder while scaling the feature size. The network concludes with a convolution layer refining the features, which are added to the interpolated frames to generate the final output.

Table 3: Performance comparisons in terms of Full-Reference IQA (DISTS [7]) and No-Reference IQA (NIQE [22] and CLIP-IQA [35]) evaluation metrics on the REDS4 [24] and Vid4 [19] datasets. The extended version of SeeClear is marked with ⋆. Red indicates the best, and blue indicates the runner-up performance (best view in color) in each group of experiments.

| Methods | Frames | REDS4 [24] | | | Vid4 [19] | | |
|---------|--------|------------|------|-----------|-----------|------|-----------|
| | | DISTS ↓ | NIQE ↓ | CLIP-IQA ↑ | DISTS ↓ | NIQE ↓ | CLIP-IQA ↑ |
| Bicubic | - | 0.1876 | 7.257 | 0.6045 | 0.2201 | 7.536 | 0.6817 |
| TOFlow [41] | 7 | 0.1468 | 6.260 | 0.6176 | 0.1776 | 7.229 | 0.7356 |
| EDVR-M [37] | 5 | 0.0943 | 4.544 | 0.6382 | 0.1468 | 5.528 | 0.7380 |
| BasicVSR [1] | 15 | 0.0808 | 4.197 | 0.6353 | 0.1442 | 5.340 | 0.7410 |
| VRT [16] | 6 | 0.0823 | 4.252 | 0.6379 | 0.1372 | 5.242 | 0.7434 |
| IconVSR [1] | 15 | 0.0762 | 4.117 | 0.6162 | 0.1406 | 5.392 | 0.7411 |
| StableSR [36] | 1 | 0.0755 | 4.116 | 0.6579 | 0.1385 | 5.237 | 0.7644 |
| ResShift [45] | 1 | 0.1432 | 6.391 | 0.6711 | 0.1716 | 6.868 | 0.7157 |
| SATeCo [6] | 6 | 0.0607 | 4.104 | 0.6622 | 0.1015 | 5.212 | 0.7451 |
| SeeClear (Ours) | 5 | 0.0762 | 4.381 | 0.6870 | 0.0947 | 5.305 | 0.7106 |
| SeeClear⋆ (Ours) | 5 | 0.0641 | 3.757 | 0.6848 | 0.0919 | 4.896 | 0.7303 |

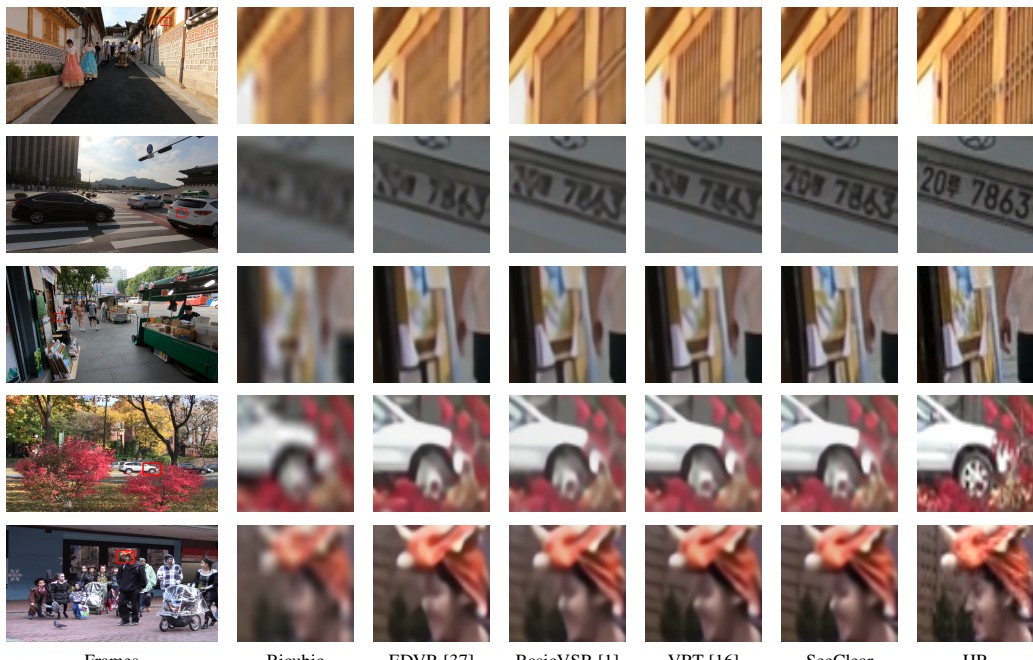

Frames  Bicubic  EDVR [37]  BasicVSR [1]  VRT [16]  SeeClear  HR

Figure 9: Visual examples of video super-resolution results by different approaches on the REDS4 and Vid4 datasets. The region in the red box is presented in the zoom-in view for comparison.

## D  Additional Experiments

We provide additional evaluation metrics and visual comparisons contrasting the existing VSR methods with our proposed SeeClear. As demonstrated in Figure 9, our proposed method successfully generates pleasing images without priors pre-trained on large-scale image datasets, showcasing sharp edges and clear details, evident in vertical bar patterns of windows and characters on license plates. Conversely, prevalent methods seemingly struggle, causing texture distortion or detail loss in analogous scenes.

We also execute several experiments focused on the degradation scheme of the diffusion process, verifying the performance of the models on the first frame, as indicated in Table 4. Compared to the straightforward diffusion process of simply adding Gaussian noise into HR frames, the combination of blurring diffusion proves beneficial in generating the high-frequency details discarded in LR

Table 4: Performance comparisons on the first frame of REDS4 among variants with different deterioration during the diffusion process and U-Net.

| | Noise | $\sigma_B^2$ | Model | PSNR ↑ | SSIM ↑ | LPIPS ↓ |
|---|---|---|---|---|---|---|
| 1 | ResShift | 0 | WaveDiff | 26.78 | 0.7960 | 0.2096 |
| 2 | ResShift | 2 | WaveDiff | 26.45 | 0.7927 | 0.2008 |
| 3 | ResShift | 3 | WaveDiff | 27.74 | 0.8047 | 0.2067 |
| 4 | ResShift | 0 | SwinUNet | 27.76 | 0.8013 | 0.2346 |
| 5 | ResShift | 2 | SeeClear | 28.04 | 0.8134 | 0.1971 |

frames. Specifically, the variation in the intensity of blur (Line 1-3) affects the fidelity and perceptual quality. Among them, there is no blurring when $\sigma_B^2 = 0$, and the greater the value of $\sigma_B^2$, the greater the blurring intensity. It can be observed there is a 0.96 dB improvement in PSNR and the value of LPIPS ranges from 0.2096 to 0.2067 with the increasement of $\sigma_B^2$.

On another note, implementing attention mechanisms in the wavelet spectrum proves more successful in uncovering valuable insights than in the pixel domain. Additionally, SeeClear introduces the alternation of spatial self-attention and channel self-attention, refining the modeling of intra and inter-frequency sub-band correlations and remarkably enhancing the quality of the generated high-resolution frames.

## E Complexity Analysis

Table 5 compares the efficiency between our proposed method and diffusion-based methods. It presents the number of parameters of different models and their inference time for super-resolving $512 \times 512$ frames from $128 \times 128$ inputs. Combining these comparative results, we draw the following conclusions: i) Compared to semantic-assisted single-image super-resolution (e.g., CoSeR [32] and SeeSR [39]), our proposed method possesses fewer parameters and higher inference efficiency. ii) In contrast to existing diffusion-based methodologies for VSR [27, 43], SeeClear is much smaller and runs faster, benefiting from the reasonable module designs and diffusion process combing patch-level blurring and residual shift mechanism.

Table 5: Complexity comparisons between different diffusion-based super-resolution methods.

| | Methods | Params | Time Steps | Infer Time (s) |
|---|---|---|---|---|
| | LDM [26] | 169.0 M | 200 | 5.21 |
| | DiffBIR [18] | 1716.7M | 50 | 5.85 |
| | ResShift [45] | 173.9M | 15 | 1.12 |
| SISR | StableSR [36] | 1409.1M | 200 | 18.70 |
| | CoSeR [32] | 2655.52M | 200 | - |
| | SeeSR [39] | 2283.7M | 50 | 7.24 |
| | PASD [42] | 1900.4M | 20 | 6.07 |
| | StableVSR [27] | 712M | 50 | 28.57 |
| VSR | MGLD-VSR [43] | 1.5B | 50 | 1.113 |
| | SeeClear (Ours) | 229.23M | 15 | 1.142 |

## F Generation Process

A video clip consisting of five frames is parallelly sampled during the inference process. These LR frames are first fed into the semantic distiller to extract semantic tokens and then corrupted by random noise as the input of the pixel condenser. The pixel condenser iteratively generates the corresponding HR counterparts from noisy LR frames under the condition of LR frames and semantic priors. The pseudo-code of the inference is depicted in the Algorithm 1.

## G Limitations and Societal Impacts

Limited by the size and diversity of the dataset, SeeClear, being solely trained on video super-resolution datasets, does not fully leverage the generative capabilities of diffusion models as efficiently as those that benefit from pre-training on large-scale image datasets such as ImageNet. While SeeClear is capable of generating sharp textures and maintaining consistent details, it falls short in restoring tiny objects or intricate structures with complete satisfaction.

**Algorithm 1:** Generation Process of SleeCear

**Input:** LR frames $I_n^{LR} \in \mathbb{R}^{N \times C \times H \times W}$; time steps $T$; and predefined schedule $\{\alpha, \eta\}$
**Output:** SR frames $I_n^{SR} \in \mathbb{R}^{N \times C \times sH \times sW}$

1   $\bar{I}_n^{LR} = \text{VRT}(I_n^{LR})$
2   $u_n^T = \text{DWT}_{2D}(\bar{I}_n^{LR}) + \epsilon, \epsilon \sim \mathcal{N}(0, \mathbf{I})$
3   —————————————— Parallel Generation within Local Video Clip ——————————————
4   **for** $m = 1$ to $\frac{N}{M}$ **do**      // $M$ refers to the number of frames in a video clip
5    **for** $t = T - 1$ to $0$ **do**
6     **if** *t=T-1* **then**
7      $O_n, P_n = \mathcal{S}(I_n^{LR}, \mathcal{V})$
8      $u_n^t = u_n^T$
9     **else**
10      $u_n^t = \text{DWT}_{2D}(\bar{I}_n^{t+1})$
11     $skip_H = [\,]$   // Array of high-frequency spectrums for skip connection
12     **for** $i = 1$ to $4$ **do**          // i refers to $i^{th}$ layer of encoder $\mathcal{E}$
13      **if** *i=1* **then**
14       $f_n^{i-1} = [I_n^t, u_n^t]$
15      **else**
16       $f_n^{i-1} = [\text{WD}(I_n^{LR}), \bar{f}_n^{i-1}]$    // WD denotes Wavelet-based DownSample
17      $\hat{f}_n^i = \mathcal{E}_i(f_n^{i-1})$
18      **if** *i ≠ 4* **then**
19       $L_n^i, H_n^i = \text{WD}(\hat{f}_n^i)$
20       $\bar{f}_n^i = L_n^i$
21       $skip_H = skip_H + H_n^i$
22      **else**
23       $\bar{f}_n^i = \hat{f}_n^i$
24     $O_m = \text{Dec}(\text{Enc}(O_n), \hat{O})$
25     **for** $j = 1$ to $3$ **do**          // j refers to $j^{th}$ layer of middle blocks $\mathcal{B}$
26      $\gamma_n^j, \beta_n^j = \mathcal{G}(P_n)$
27      $f_n^j = (\bar{f}_n^{j-1} \bigodot \gamma_n^j + \beta_n^j) + \bar{f}_n^{j-1}$
28      $\hat{f}_n^j = \text{CA}(f_n^j, O_m, O_m)$
29      $\bar{f}_n^j = \mathcal{B}_j(\text{MFSA}(O_c \cdot \hat{f}_n^j))$
30     $feats = [\,]$         // Array of multi-scale features for CaTeGory
31     **for** $k = 1$ to $4$ **do**          // k refers to $k^{th}$ layer of decoder $\mathcal{D}$
32      **if** *k≠4* **then**
33       $f_n^k = \text{WU}([\bar{f}_n^{k-1}, skip_H[-k]])$ // WU denotes Wavelet-based UpSample
34      $\gamma_n^k, \beta_n^k = \mathcal{G}(P_n)$
35      $\hat{f}_n^k = (f_n^{k-1} \bigodot \gamma_n^k + \beta_n^k) + \bar{f}_n^{k-1}$
36      $\widetilde{f}_n^k = \text{CA}(\hat{f}_n^k, O_c, O_c)$
37      $\ddot{f}_n^k = \mathcal{D}_k(\text{MFSA}(O_c \cdot \widetilde{f}_n^k))$
38      $\bar{f}_n^k = \text{CA}(\ddot{f}_n^k, C_m, T_m)$
39      $feats = feats + \bar{f}_n^k$
40     $I_n^t = \text{IDWT}_{2D}(\bar{f}_n) + \bar{I}_n^{LR}$
41     $\bar{I}_n^t = \alpha_t \mu(I_n^t, \bar{I}_n^{t+1}) + \eta_t \epsilon$
42     —————————————— Update CaTeGory for Global Video Consistency ——————————————
43     **if** *t=0* **then**
44      $C_{m+1}, T_{m+1} = \mathcal{M}(C_m, T_m, feats)$
45      $I_n^{SR} = I_n^t$

In addition, compared to synthetic datasets, videos captured in real-world scenarios display more complex and unknown degradation processes. Although real-world VSR is garnering significant attention, it remains an unexplored treasure and a steep summit to conquer for diffusion-based VSR, including SeeClear.

Significantly, substituting an Autoencoder with a traditional wavelet transform lessens the computational burden while ensuring the preservation of the original input information. Nevertheless, in the process of inverse wavelet transform, the spatio-temporal information within videos is not delved into further, leading to subpar outcomes. Meanwhile, some methods develop trainable wavelet-like transformations based on the lifting scheme, allowing for end-to-end training of the whole network. Such a schema presents a promising direction for future research by potentially boosting model performance.

As for societal impacts, similar to other restoration methods, SeeClear may bring privacy concerns after restoring blurry videos and lead to misjudgments if used for medical diagnosis and intelligent security systems, etc.

