# OpenReview forum: "SeeClear: Semantic Distillation Enhances Pixel Condensation for Video Super-Resolution"
_NeurIPS.cc/2024/Conference — NeurIPS 2024 poster_

### Official Review · Reviewer_ERJ1 · 2024-07-12

**Soundness:** 2
**Presentation:** 3
**Contribution:** 3
**Rating:** 6
**Confidence:** 4

**Summary:**

The authors propose SeeClear for Video Super-Resolution (VSR). SeeClear is a diffusion-based method that improves restoration performance by introducing semantic priors. The authors design an Instance-Centric Alignment Module (InCAM) and Channel-wise Texture Aggregation Memory (CaTeGory) to utilize semantic information effectively. Comparisons on multiple datasets demonstrate that the proposed method achieves state-of-the-art performance.

**Strengths:**

1. The paper introduces semantic priors to achieve spatial modulation and temporal correlation, improving diffusion-based VSR performance. This idea is both reasonable and effective.
2. The authors design the Instance-Centric Alignment Module (InCAM) to align using semantic information, avoiding pixel inconsistencies and being well-suited for diffusion models.
3. Additionally, the authors propose the Channel-wise Texture Aggregation Memory (CaTeGory) to transfer semantic information between different frames.
4. Comparisons with state-of-the-art methods demonstrate the effectiveness of the proposed method.
5. The paper is well-organized, with clear and aesthetically pleasing layouts, figures, and tables.

**Weaknesses:**

1. The method uses pre-trained models to extract semantic information, introducing significant additional computation, which limits the method's applicability. Meanwhile, the paper lacks comparisons of complexity and parameter counts.
2. The method lacks experimental support for some critical hyperparameters, such as the choice of k in InCAM and the number of frames used in SR.
3. The paper proposes using wavelet transform to improve UNet but lacks experimental justification for why simple downsampling and upsampling wouldn't be more efficient.
4. Figure 1, while aesthetically pleasing, is challenging to understand. It would be better to clearly explain the network structure (e.g., Figure 8) and the inference process.

**Questions:**

1. Why do the comparison methods in Table 1 use different numbers of frames? If the same frame is used, what is the performance like?
2. In the ablation study (model 2, Table 2), how to use semantic conditions without MFSA, InCAM, and CaTeGory?
3. In InCAM, what is the value of k in top k, and how is it determined? Is there experimental support for this choice?
4. Others see weaknesses.

**Limitations:**

The authors discuss the method's limitations and societal impacts.

---

> ### Author Rebuttal · Authors · 2024-08-07
>
> > Q1. Why do the comparison methods in Table 1 use different numbers of frames? If the same frame is used, what is the performance like?
>
> The selection of numbers of frames for training depends on the architecture, such as sliding-window-based (e.g., EDVR-M) and recurrent-based (e.g., IconVSR) methods. The longer temporal information the model can access during training, the better the performance could be generally attained [1, 2]. Nevertheless, the increase in the number of frames will also lead to huge computational overhead. Therefore, we propose to parallelly align inter-frame within the short clip and maintain long-term temporal consistency via additional texture memory.
>
> [1] Shuwei Shi, Jinjin Gu, Liangbin Xie, Xintao Wang, Yujiu Yang, and Chao Dong. Rethinking Alignment in Video Super-Resolution Transformers. In NeurIPS, 2022.
>
> [2] Kelvin C.K. Chan, Shangchen Zhou, Xiangyu Xu, and Chen Change Loy. Investigating Tradeoffs in Real-World Video Super-Resolution. In CVPR, 2022.
>
> > Q2. In the ablation study (model 2, Table 2), how to use semantic conditions without MFSA, InCAM, and CaTeGory?
>
> The semantic conditions are utilized for the generation of each frame via cross-attention, enabling the SeeClear to be cognizant of the content to be generated. Specifically, the query and key/value are respectively projected from features of SR and instance-centric semantic embedding of segmentation. It embeds the semantic priors into the pixel-level features, similar to the position embedding.
>
> > Q3. In InCAM, what is the value of k in top k, and how is it determined? Is there experimental support for this choice?
>
> On the one hand, the resolution of the LR frames constrains the value range of k. On the other hand, there is saturation in the choice of k, as shown in Table 3 of the rebuttal document. As long as the instances and backgrounds in the frames can be retrieved completely, a larger k won't lead to performance improvement while increasing the computational cost of the network.
>
> > Q4. The method uses pre-trained models to extract semantic information, introducing significant additional computation, which limits the method's applicability. Meanwhile, the paper lacks comparisons of complexity and parameter counts.
>
> Albeit an additional segmentation network is introduced, the auto-encoder is replaced by DWT simultaneously, making it possible to increase the network scale and improve the performance. Additionally, the option of the segmentation network is highly flexible, as long as it is transformer-based architecture. Especially with the continuous development of segmentation tasks in recent years, it is highly beneficial for performance gain of SeeClear. The semantic-assisted generative super-resolution comes forth in recent advanced works [1, 2]. Some of them not only require additional training for the semantic extraction network but also add ControlNet as a condition control mechanism, bringing more parameters. To further demonstrate the effectiveness and efficiency of the SeeClear, we also compare it with other representative diffusion-base SR methods in terms of the number of parameters and inference time, please see Q2 of the global response for more information.
>
> [1] Haoze Sun, Wenbo Li, Jianzhuang Liu, Haoyu Chen, Renjing Pei, Xueyi Zou, Youliang Yan, and Yujiu Yang. CoSeR: Bridging Image and Language for Cognitive Super-Resolution. In CVPR, 2024.
>
> [2] Rongyuan Wu, Tao Yang, Lingchen Sun, Zhengqiang Zhang, Shuai Li and Lei Zhang. SeeSR: Towards Semantics-Aware Real-World Image Super-Resolution. In CVPR, 2024.
>
> > Q5. The paper proposes using wavelet transform to improve UNet but lacks experimental justification for why simple downsampling and upsampling wouldn't be more efficient.
>
> Please see the answer to Q1 of the global response.
>
> > Q6. Figure 1, while aesthetically pleasing, is challenging to understand. It would be better to clearly explain the network structure (e.g., Figure 8) and the inference process.
>
> The Figure 1 illustrates the two main networks (i.e., semantic distiller for semantic extraction and pixel condenser for denoising) as a conceptual sketch, rather than a main figure. To further demonstrate the network architecture of the network and the its connection with the devised components, we supplement detailed illustration and explains, as you seen in the Figure 8 and subsequent text in the **Section B** of supplemental materials.
>
> A video clip consisting of five frames is parallelly sampled during the inference process. These LR frames are first fed into the semantic distiller to extract semantic tokens and then corrupted by random noise as the input of the pixel condenser. The pixel condenser iteratively generates the corresponding HR counterparts from noisy LR frames under the condition of LR frames and semantic priors. The specific information flow of the pixel condenser is elaborated in Section B of supplemental materials. Due to the space limitation, we would like to supplement the pseudo-code of the inference in the final edition.

---

> > ### Comment · Reviewer_ERJ1 · 2024-08-12
> > **After Rebuttal**
> >
> > Thanks for the rebuttal. The authors provide extensive experiments to demonstrate the effectiveness of the proposed method.
> >
> > I also read the comments of other reviewers. The main concerns are about the lack of experiments, and I think the authors' experiments addressed these issues to some extent.
> >
> > Overall, the authors address my concerns, and I am willing to raise my score to 6.

---

### Official Review · Reviewer_srYj · 2024-07-12

**Soundness:** 2
**Presentation:** 3
**Contribution:** 3
**Rating:** 5
**Confidence:** 4

**Summary:**

The paper introduces a novel video super-resolution framework leveraging semantic distillation to enhance pixel condensation in diffusion-based models. SeeClear addresses stochastic fluctuations by using a Semantic Distiller and a Pixel Condenser to extract and upscale semantic details from LR frames. The framework includes an Instance-Centric Alignment Module and a Channel-wise Texture Aggregation Memory to improve temporal consistency and visual quality. Experimental results demonstrate SeeClear's superiority over state-of-the-art diffusion-based VSR techniques.

**Strengths:**

- The combination of semantic distillation and pixel condensation is novel and effectively addresses the challenges of maintaining detail consistency across frames in diffusion-based VSR.
- The Instance-Centric Alignment Module (InCAM) and Channel-wise Texture Aggregation Memory (CaTeGory) significantly improve short-term and long-term temporal coherence.
- The paper provides extensive experiments to demonstrate SeeClear's advantages over sotas across multiple benchmarks.

**Weaknesses:**

- Lack of computation analysis. Diffusion-based methods are often criticized for unbearable inference time, so it would be better to list params, runtime, and FLOPs/MACs for a fair comparison.
- Lack of an ablation study on the wavelet transform which is introduced in Section 3.1.
- Table 2 is incomplete, making it difficult to assess the effect of the CaTeGory.
- The Other baselines such as VRT and IconVSR are also evaluated on Vimeo-90K-T and UDM10 datasets. Could you complete it for a fair comparison?
- Figure 7 needs more explanation.

**Questions:**

Diffusion-based models usually show poor performance on PSNR (e.g., StableSR and Reshift), but SeeClear demonstrates a significant improvement. Could you analyze which parts of SeeClear contribute to this improvement?
Please refer to the weaknesses part above.

**Limitations:**

The authors have largely addressed their limitations.

---

> ### Author Rebuttal · Authors · 2024-08-07
>
> > Q1. Diffusion-based models usually show poor performance on PSNR (e.g., StableSR and Reshift), but SeeClear demonstrates a significant improvement. Could you analyze which parts of SeeClear contribute to this improvement? Please refer to the weaknesses part above.
>
> Diffusion-based image super-resolution acquires remarkable perceptual quality thanks to the generative capabilities of the models, yet the consistency with low-resolution images is somewhat overlooked. However, the highly correlated content within adjacent frames of a video supplies more references and constraints for the generation of individual frames. Coupled with InCAM, which activates and associates semantic-related pixels among adjacent frames, it's possible to enhance the perceptual quality of the reconstructed video and strengthen the constraints on fidelity, as shown in Table 2 of the paper. Similar experimental results can also be observed in the comparative method SATeCo. Besides, the pixel condenser employs wavelet transform to change the resolution of features and continually reinject significant information into downsampled features from LR frames, guaranteeing the content fidelity. And the split high-frequency components of LR frames are transmitted to the decoder via skip connections, guiding the generation of details and textures. Additionally, some works [1, 2] also provide theoretical support for this experimental result. In conclusion, we hold the belief that the constraints of multiple frames of the video and the appropriate module designs jointly constrain the solution space of the model.
>
> [1] Theo Adrai, Guy Ohayon, Michael Elad, and Tomer Michaeli. Deep Optimal Transport: A Practical Algorithm for Photo-realistic Image Restoration. In NeurIPS, 2023.
>
> [2] Dror Freirich, Tomer Michaeli, and Ron Meir. A Theory of the Distortion-Perception Tradeoff in Wasserstein Space. In NeurIPS, 2021.
>
> > Q2. Lack of computation analysis.
>
> Please see the answer to Q2 of the global response.
>
> > Q3. Lack of an ablation study on the wavelet transform.
>
> Please see the answer to Q1 of the global response.
>
> > Q4. The Other baselines such as VRT and IconVSR are also evaluated on Vimeo-90K-T and UDM10 datasets. Could you complete it for a fair comparison?
>
> Due to the lack of diffusion-based VSR evaluated onVimeo-90K-T and UDM10 datasets, we can only just provide the quantitative result generated via our proposed method.
>
> |Datasets|	PSNR  $\uparrow$|	SSIM  $\uparrow$|	LPIPS  $\downarrow$|
> |:---:|:---:|:---:|:---:|
> |Vimeo-90K-T (BI)	|37.64|0.9503|0.0982|
> |UDM10 (BD)|	39.72|	0.9675|	0.0609|
>
>
> > Q5. Figure 7 needs more explanation.
>
> Please refer to the answer to the Q3 of the global response.

---

> > ### Comment · Reviewer_srYj · 2024-08-13
> >
> > Thank you for your rebuttal. I raised my rating to borderline accept.

---

### Official Review · Reviewer_vLVk · 2024-07-13

**Soundness:** 3
**Presentation:** 3
**Contribution:** 3
**Rating:** 6
**Confidence:** 4

**Summary:**

This paper presents a diffusion-based video super-resolution method, and proposes Instance-Centric Alignment Module and Channel-wise Texture Aggregation Memory. The former leverages a pre-trained open-vocabulary segmentation model (i.e., OpenSeeD), which is utilized to perform alignment within video clips by modulating the spatial and temporal features. The latter leverages channel-wise attention and memory mechnism to better super-resolve the video frames. The results on publich benchmarks indicate that the proposed method achieves state-of-the-art perceptual performance.

**Strengths:**

1. The proposed method achieves state-of-the-art perceptual results on REDS4 and Vid4.

**Weaknesses:**

Although the proposed method achieves promising results on the public benchmarks, there are some concerns that greatly affect the rating of this paper.

1. The presentation of the method needs to be improved. The readability of the paper is unsatisfactory. The technical details and the rationale behind it is not clearly described and explained.
(a) The main figure (Figure 1) is ambiguous. It is hard to understand the workflow of the framework based on this figure. It is also hard to see the connection among different modules.
(b) In the abstract, what is the "conditional video generation" (L6)? I do not see any pre-trained conditional video generation module in the described method. Maybe it should be rephrased.
(c) In L206-207, what is the role of "randomly initialized tokens"? And what is specific role of the encoder-decoder module?
(d) In L187-188, are the "semantic tokens" actually text embeddings ? What is the difference?
(e) In L223, how to divide channels into different groups and what is rationale behind it?
(f) It is hard to understand Eq. (16), (17) and (18). From (17) and (18), it seems T_j is used to calculate itself, which is confusing.
(g) The choice of mathematical notations is sub-optimal and confusing.
(h) In L149, I think the "belta_t" should be "yita_t".

2. The novelty of this paper is limited.
(a) Some of the modules are based on existing methods. For example, the way of introducing semantic features is similar to SFT (but no comparison in the paper); the multi-frame self attention is from [21].
(b) The proposed blurring ResShift is a modification version based on ResShift, but the rationale behind it is not fully explained. Also, there is no direct ablation.

3. The comparison with other related methods are not thorough.
(a) The authors should explicitly compare with ResShift [33], since residual shifting technique is also exploited (but no citation in L48). Also, there is no comparison with it in Sec. 2.2.
(b) The authors should compare with Upscale-A-Video [36], another diffusion-based video super-resolution method. Also, it is recommended to compare the performance of [36].
(c) The authors should compare with SFT[28], another method also leveraging semantic segmentation information.

4. The proposed method is not fully ablated. There is no direct ablation for exploitation of DWT and blurring resshift.

5. Some of the statements could be inappropriate.
(a) In L35-36, I think it is hard to reach the given conclusion from [8]. Please elaborate.
(b) The naming of "semantic distiller" could be inappropriate. The pre-trained semantic segmentation model is directly leveraged and frozen. I don't see any distillation.

**Questions:**

1. In the abstract, what is the "conditional video generation" (L6)? I do not see any pre-trained conditional video generation module in the described method.
2. In L206-207, what is the role of "randomly initialized tokens"? And what is specific role of the encoder-decoder module?
3. In L187-188, are the "semantic tokens" actually text embeddings ? What is the difference?
4. In L223, how to divide channels into different groups and what is rationale behind it?
5. It is hard to understand Eq. (16), (17) and (18). From (17) and (18), it seems T_j is used to calculate itself, which is confusing. Please elaborate.
6. In InCAM, the way of introducing semantic features seems similar to SFT. Please compare with it and illustrate the significance of the proposed module.
7. Please provide comparison with the following related methods, and illustrate the novelty of the proposed modules.
(a) The necessity and rationale of blurring ResShift, and the ablation study.
(b) Upscale-A-Video [36]. And please compare performance with it quantitatively.
8. In L35-36, I think it is hard to reach the given conclusion from [8]. Please elaborate.

**Limitations:**

The authors have adequately addressed the limitations in Section D.

---

> ### Author Rebuttal · Authors · 2024-08-07
>
> Thanks for your careful reading and detailed comments. We will rectify some confusing statements and formulas in the subsequent edition. Nevertheless, we deem it necessary to highlight our novelty and restate the proposed method. Different from the **text** in the realm of T2I and **segmentation mask** in existing generative super-resolution, we pioneer the exploration of the utilization of temporal information based on the semantic similarity for diffusion-based VSR. The instance-centric semantic embeddings and channel-wise semantics are employed to determine the conditional pixels with high quality and semantic similarity from adjacent frames and long-term temporal information, enhancing the generative quality and temporal consistency.
>
> > Q1. What is the "conditional video generation"?
>
> As depicted in Figure 1 of the rebuttal document, SeeClear consists of a forward diffusion process and a reverse process for VSR. In the diffusion process, patch-level blurring and residual shift mechanism are integrated to degrade HR frames based on the handcrafted time schedule. During the reverse process, a transformer-based network for open vocabulary segmentation and a U-Net are employed for iterative denoising. The former is responsible for extracting semantic embeddings related to instances from LR videos, similar to the process of distillation in physics, and is therefore named the semantic distiller. The latter is utilized to filter out interfering noise and retain valuable information from low-quality frames, similar to the condensation process. All of them are tailored for image processing, and SeeClear takes diverse semantic embeddings as conditions to enable the network to be aware of the generated content and determine the aligned pixels from adjacent frames for the temporal consistency of the whole video. More details of U-Net for denoising are elaborated in **Section B** of the supplementary material.
>
> > Q3 & Q4. Are the "semantic tokens" actually text embeddings?  How to divide channels into different groups?
>
> The conditions in SeeClear refer to instance-centric semantic embeddings, semantics within the channel dimension and discriminative textures, rather than text or segmentation mask. The semantic distiller comprises three components, i.e., backbone, encoder and decoder. The encoder takes multi-scale features from backbone as input and models the relationship among pixels via self-attention. Randomly initialized tokens serve as the input of the decoder together with the dense features yielded by the encoder, which integrate instance-related semantic information through cross-attention. Eventually, those semantic tokens are utilized to predict classes, bounding boxes, and masks by simple modules like MLP.
>
> Besides, we dig abundant semantics within the channel dimension via CaTeGory. For example, different channels response to edges/corners in shallow layer, and deeper layers further extract complex patterns by combining the information of different channels in the previous layer. CaTeGory consists of semantic and texture embeddings, which are zero-initialized as parameters of network. Throughout the training process, semantic embeddings iteratively aggregate the channel semantic from instance-centric semantics via cross-attention, and texture embeddings formulate representative textures from the multi-scale features of the U-Net. To establish the one-to-one correspondence between channel-wise semantics and textures, CaTeGory correlates semantic and texture embeddings through element-wise multiplication.
>
> > Q2 & Q6. What is the role of "randomly initialized tokens"? And what is specific role of the encoder-decoder module? In InCAM, the way of introducing semantic features seems similar to SFT.
>
> To effectively utilize these semantic conditions, InCAM is inserted before the spatial self-attention of U-Net, aiming at performing inter-frame alignment in the semantic space implicitly. Distinct from SFT, which merely generates coefficients for feature modulation based on class priors derived from the segmentation mask, InCAM utilizes two types of semantic priors, i.e., semantic embeddings and dense features. Among them, dense features and those of VSR are jointly employed for the generation of modulation coefficients, enabling the modulated features to lie in the domain between the segmentation and the super-resolution, narrowing the gap between semantic priors and pixel features. Moreover, the semantic embeddings are not only used as conditions for the generation of each frame, but also can locate pixels related to specific instances within the frame. Therefore, InCAM incorporates the instance encoder-decoder (L206-207) to yield clip-wise semantics based on the instance-centric semantics of adjacent frames, which associate the same instances and align the semantically related pixels between adjacent frames. The random initialized tokens input in the decoder will serve as clip-wise semantics after the iterative interaction along temporal axis. From this perspective, InCAM also possesses the ability of inter-frame alignment that SFT lacks.
>
> > Q5. It is hard to understand Eq. (16), (17) and (18). From (17) and (18). Please elaborate.
>
> Please see the official comment.
>
> > Q7 & Q8. Please provide comparison with the following related methods. I think it is hard to reach the given conclusion from [8]. Please elaborate.
>
> Due to the unavailability of Upscale-A-Video’s code and the relevant datasets (e.g., REDS30), a fair comparison cannot be conducted. But we are willing to carry out the comparison in our subsequent versions once the code and related datasets are released. While SFT is an image-level SR method and lacks quantitative results, thus we add other representative methods in the response to Q2 for Reviewer NXC1.
>
> Please refer to the answer to Q3 in the global response for the elaboration of the given conclusion.

---

> ### Author Response · Authors · 2024-08-07
> **Reformulation of Eq. (16)-(18).**
>
> We reformulate Eq. (16)-(18) as follows:
>
> $$
> \left(C_j, T_j\right)=\mathcal{M}\left(\bar{C}_j, \bar{T}_j,\left\\{\bar{F} _{i, k}\right\\} _{k=1}^4\right)
> $$
>
> $$
> \widehat{T}_j=\bar{C}_j \times \bar{T}_j
> $$
>
> $$
> T_j=\operatorname{SA}\left(\operatorname{CA}\left(\widehat{T}_j,\left\\{\bar{F} _{i, k}\right\\} _{k=1}^4\right)\right)
> $$
>
> where $\bar{C}_j$ and $\bar{T}_j$ respectively denote zero-initialized channel-wise semantics and textures of j-th group. $\widehat{T}_j$ is the texture embedding that possesses one-to-one correspondence with semantic embedding via matrix multiplication.

---

> ### Comment · Reviewer_vLVk · 2024-08-11
> **Response to the rebuttal**
>
> Q3: The answer is still unclear. I still do not unstand the meaning of "semantic tokens" (O) and what are the relation with the visual features (F).
> Q4: I do not see any direct answer to the question of how to divide channels.
> Q8: The answer to Q3 seems to have no relation with Q8.
>
> I would like to maintain my current rating.

---

> > ### Author Response · Authors · 2024-08-12
> >
> > Thanks for your valuable comments. We provide more materials and details for understanding semantic tokens and the way to divide channels. Besides, there’re sufficient theories and experimental results to demonstrate the necessity of combination of blurring and additive noise in the diffusion process for SR, i.e., Blurring ResShift. We have also provided detailed ablation experiments and explanations to validate the significance of the proposed Blurring ResShift during the rebuttal period.
> >
> > > Q3: The answer is still unclear. I still do not unstand the meaning of "semantic tokens" (O) and what are the relation with the visual features (F).
> >
> > The same terms are employed in previous works [3, 5]. Encoder-decoder based transformer architecture for segmentation has been prevalently utilized in recent years, and the concept of semantic token is blatantly simple to understand and widely accepted. We firmly don't think it absolutely constitute a reason for rejection.
> >
> > As clearly stated in the rebuttal, transformer-based segmentation network [1, 2, 3, 4] requires **two types of input, i.e., images and learnable tokens**, which is well-understood in the realm of visual tasks and has also been applied in numerous image segmentation tasks.  The encoder of the transformer extracts multi-scale features from images via self-attention (Refer to Fig. 2 and its caption in [1]), yielding visual features (F). Meanwhile, the decoder of the transformer establishes relationships between learnable tokens and multi-scale features (Refer to “box embeddings”, “encoded image” and “Resnet features” of Fig. 8 in [1]), generating **semantic tokens (O) that can retrieve pixels related to instances and background from visual features** (Refer to “object queries” of Fig. 2 in [1]). They are combined to produce the final segmentation mask through MLP or element-wise multiplication.
> >
> > [1] Carion N, Massa F, Synnaeve G, et al. End-to-end object detection with transformers[C]//**European Conference on Computer Vision (ECCV)**. Cham: Springer International Publishing, 2020: 213-229.
> >
> > [2] Cheng B, Misra I, Schwing A G, et al. Masked-attention mask transformer for universal image segmentation[C]//Proceedings of the **IEEE/CVF Conference on Computer Vision and Pattern Recognition (CVPR)**. 2022: 1290-1299.
> >
> > [3] Zou X, Dou Z Y, Yang J, et al. Generalized decoding for pixel, image, and language[C]//Proceedings of the **IEEE/CVF Conference on Computer Vision and Pattern Recognition (CVPR)**. 2023: 15116-15127.
> >
> > [4] Li X, Ding H, Yuan H, et al. Transformer-based visual segmentation: A survey[J]. **IEEE Transactions on Pattern Analysis and Machine Intelligence (T-PAMI)**, 2024.
> >
> > [5] Zhang H, Li F, Zou X, et al. A simple framework for open-vocabulary segmentation and detection[C]//Proceedings of the **IEEE/CVF International Conference on Computer Vision (CVPR)**. 2023: 1020-1031.
> >
> > > Q4: I do not see any direct answer to the question of how to divide channels.
> >
> > The direct answer has been elaborated in the second paragraph beginning with “we dig abundant”，but we’d like to explain it in a simple way. The cluster of channel relies on two zero-initialized parameters of network, i.e., channel-wise semantics ($C_j$) and textures ($T_j$). The semantic parameters initially model the relationship with channels of instance-centric tokens ($O_i$) via cross-attention along channel dimension instead of vanilla spatial cross-attention, which is formulated as:
> >
> > $$
> > Q=C_j W^Q,K=O_i W^K,V=O_i W^V
> > $$
> > $$
> > A=\operatorname{softmax}((Q^T K)/\sqrt{d})
> > $$
> > $$
> > \widetilde{C}_j =AV+C_j
> > $$
> >
> > After stacked cross-attention and residual connection, the semantics contained in channels of instance-centric semantic tokens are resembled into semantic parameters, thus grouping different channel-wise semantics. Then, the texture parameter is combined with semantic parameter via matrix multiplication. It appoints each channel semantic to a corresponding texture. The texture parameter assembles valuable textures from multi-scale features via cross-attention, similarly to the above process. The motivation and technical details are elaborated, we think that you should read our paper and rebuttal more attentively.
> >
> > > Q5: Where is the "official comments"?
> >
> > You may not be able to access the reformulation of Eq. (16)-(18) provided during the rebuttal period due to system. We’d like to exhibit the same contents here.
> >
> > We reformulate Eq. (16)-(18) as follows:
> >
> > $$
> > \left(C_j, T_j\right)=\mathcal{M}\left(\bar{C}_j, \bar{T}_j,\left\\{\bar{F} _{i, k}\right\\} _{k=1}^4\right)
> > $$
> >
> > $$
> > \widehat{T}_j=\bar{C}_j \times \bar{T}_j
> > $$
> >
> > $$
> > T_j=\operatorname{SA}\left(\operatorname{CA}\left(\widehat{T}_j,\left\\{\bar{F} _{i, k}\right\\} _{k=1}^4\right)\right)
> > $$
> >
> > where $\bar{C}_j$ and $\bar{T}_j$ respectively denote zero-initialized channel-wise semantics and textures of j-th group. $\widehat{T}_j$ is the texture embedding that possesses one-to-one correspondence with semantic embedding via matrix multiplication.

---

> > > ### Comment · Reviewer_vLVk · 2024-08-12
> > > **Response to the authors**
> > >
> > > Thank you for your detailed response.
> > > I have actually misunderstood some technical details, and the response helps a lot.
> > > The technical soundness and the novelty can be rated with a higher score.
> > > I will raise my rating to "weak accept".

---

> > ### Author Response · Authors · 2024-08-12
> >
> > > Q7: Still no further explanation on the significance of the proposed blur ResShift.
> >
> > We conducted comprehensive ablation experiments of Blurring ResShift in the complementary materials and presented it in the attached rebuttal document again. As shown in Table 2 of the rebuttal document, the variation in the intensity of blur (Line 1-3) affects the fidelity and perceptual quality. Among them, there is no blurring when $\sigma_B^2=0$, and the greater the value of $\sigma_B^2$, the greater the blurring intensity. It can be observed there is a 0.96 dB improvement in PSNR and the value of LPIPS ranges from 0.2096 to 0.2067 with the increasement of $\sigma_B^2$. Although the PSNR if not good enough when $\sigma_B^2=2$, the incorporation of SeeClear enables it to achieve a double-win in terms of both PSNR and LPIPS, such as the alternation of spatial self-attention and channel-wise self-attention, wavelet-based downsampling, etc. We believe that the sufficient experiments demonstrate the effectiveness and necessity of Blurring ResShift.
> >
> > > Q8: The answer to Q3 seems to have no relation with Q8.
> >
> > To answer the question you previously asked in Q8, we offer explanations from two  perspectives. Firstly, there are less high-frequency information in LR frames compared to HR frames [1], and further advancing DDPMs requires finer high-frequency details prediction [2]. Besides, diffusion-based SR methods tend to generate rather different outputs for the same LR image due to inherent stochasticity. To address these issues, an ideal diffusion process can follow a similar distribution in the low-frequency components and introduce more stochasticity in the high-frequency spectra. In other words, the network is forced to focus more on the feature extraction and modeling in the high-frequency bands, thus generating more pleasant details and textures with consistent structures [3, 4]. Therefore, we visualize the Power Spectral Density of terminated states generated by different diffusion processes in the rebuttal document, as illustrated in Figure 2. Based on the lack of high-frequency information and importance of fidelity, sole blurring or additive noise is not enough for SR, as they respectively eliminate too much low-frequency information and introduces interfered high-frequency information. Secondly, the ablation experiments on the intensity of blurring also validate the significance of combination of blurring and noise, which has been elaborated above. In a nutshell, there are sufficient theories and experimental evidence to support the conclusion.
> >
> > [1] Wang X, Chai L, Chen J. Frequency-Domain Refinement with Multiscale Diffusion for Super Resolution[J]. arXiv preprint arXiv:2405.10014, 2024.
> >
> > [2] Moser B, Frolov S, Raue F, et al. Waving goodbye to low-res: A diffusion-wavelet approach for image super-resolution[J]. arXiv preprint arXiv:2304.01994, 2023.
> >
> > [3] Sun L, Wu R, Zhang Z, et al. Improving the stability of diffusion models for content consistent super-resolution[J]. arXiv preprint arXiv:2401.00877, 2023.
> >
> > [4] Xiao X, Li S, Huang L, et al. Multi-Scale Generative Modeling in Wavelet Domain[J].

---

### Official Review · Reviewer_NXC1 · 2024-07-16

**Soundness:** 2
**Presentation:** 2
**Contribution:** 2
**Rating:** 3
**Confidence:** 5

**Summary:**

The paper presents a framework for video super-resolution (VSR) that improves temporal coherence and high-resolution detail generation. The proposed method, SeeClear, integrates a Semantic Distiller and a Pixel Condenser to extract and upscale semantic details from low-resolution frames. The framework employs an Instance-Centric Alignment Module (InCAM) and Channel-wise Texture Aggregation Memory (CaTeGory) to enhance inter-frame coherence and incorporate long-standing semantic textures. The methodology also introduces a blurring diffusion process with the ResShift mechanism to balance sharpness and diffusion effects. Experimental results show that SeeClear outperforms state-of-the-art diffusion-based VSR techniques in terms of perceptual quality and temporal consistency.

**Strengths:**

1. The SeeClear framework introduces a combination of semantic distillation and pixel condensation, which significantly enhances video super-resolution.
2. The Instance-Centric Alignment Module (InCAM) and Channel-wise Texture Aggregation Memory (CaTeGory) improve the temporal coherence of the generated high-resolution videos.
3. The integration of blurring diffusion with the ResShift mechanism effectively balances sharpness and diffusion, leading to high-quality detail generation.

**Weaknesses:**

1. While the method demonstrates robust restoration capabilities, it may still struggle with accurately restoring tiny objects or intricate structures, especially under severe degradation conditions.
2. The method has been tested primarily on specific benchmark datasets. Its performance in real-world applications, where video degradation processes are more varied and unpredictable, remains to be thoroughly evaluated.
3. The experiments are not sufficient and should be improved.

**Questions:**

1. The performance of the proposed method is not significant. In Table 1, the improvement is very marginal or is worse than other methods. Moreover, in Figure 4, the generated texture is comparable to other methods.

2. It would be better to compare more methods. (a) Transformer-based (e.g., VSR Transformer) or RNN-based method (e.g., BasicVSR++); (b) Diffusion-based image restoration methods (e.g., DDRM, DDNM, DeqIR, etc). (c) Compare methods trained with more frames (e.g., VRT-16).

3. The authors should compare the efficiency, including model size, training/inference time and FLOPs. The efficiency comparisons can demonstrate the effectiveness of the proposed method.

**Limitations:**

Please refer to the details above.

---

> ### Author Rebuttal · Authors · 2024-08-07
>
> > Q1. The performance of the proposed method is not significant. In Table 1, the improvement is very marginal or is worse than other methods. Moreover, in Figure 4, the generated texture is comparable to other methods.
>
> For the sake of fair comparison, SeeClear is trained only on five frames and achieves PSNR/SSIM comparable to those models trained on longer sequences, along with the best LPIPS. SeeClear also acquires the superior PSNR on the Vid4 dataset, surpassing the runner-up [1] by **0.36 dB**. It is worth highlighting that the novelty of SeeClear lies in its pioneering exploration of semantic tokens in the activation and association of semantic-related pixels across frames during the inter-frame alignment of diffusion-based VSR, rather than merely the improvement of performance metrics. The instance-centric semantic tokens directly extracted from frames not only stimulate the generation potential of the diffusion model but also avoid the cross-modal alignment between text and images. Besides, the inter-frame related conditional pixels determined by semantic similarity equips the diffusion-based VSR model with cognitive ability along the temporal dimension, reducing the interference of irrelevant pixels and motion estimation errors and enhancing the quality of the reconstructed videos. Moreover, the restoration of tiny objects or intricate structures and real-world VSR demands specific training datasets [2, 3, 4], which are not coincident with the configurations and benchmarks of prevailing VSR. These issues have been thoroughly discussed in **Section D** of the supplementary materials and are considered as our future undertaking.
>
> [1] Zhikai Chen, Fuchen Long, Zhaofan Qiu, Ting Yao, Wengang Zhou, Jiebo Luo, and Tao Mei. Learning spatial adaptation and temporal coherence in diffusion models for video super-resolution. In CVPR, 2024.
>
> [2] Fanghua Yu, Jinjin Gu, Zheyuan Li, Jinfan Hu, Xiangtao Kong, Xintao Wang, Jingwen He, Yu Qiao, and Chao Dong. Scaling up to excellence: Practicing model scaling for photo-realistic image restoration in the wild. In CVPR, 2024.
>
> [3] Yinhuai Wang, Jiwen Yu, and Jian Zhang. Zero-shot image restoration using denoising diffusion null-space model. In ICLR, 2023.
>
> [4] Jiezhang Cao, Yue Shi, Kai Zhang, Yulun Zhang, Radu Timofte, and Luc Van Gool. Deep equilibrium diffusion restoration with parallel sampling. In CVPR, 2024.
>
> > Q2. It would be better to compare more methods. (a) Transformer-based (e.g., VSR Transformer) or RNN-based method (e.g., BasicVSR++); (b) Diffusion-based image restoration methods (e.g., DDRM, DDNM, DeqIR, etc). (c) Compare methods trained with more frames (e.g., VRT-16).
>
> We add more methods for comparison, including RNN-based (BasicVSR++), Transformer-based (RVRT, VRT) and Diffusion-based (DDNM) methods. All these methods except DDNM are trained with more frames. It can be observed that more complex architectures (e.g., second-order grid propagation) and longer sequence used for training can significantly enhance the quality of reconstructed video but inevitably result in substantial computational overload.
>
> |Methods|Frames|PSNR $\uparrow$|SSIM  $\uparrow$|LPIPS  $\downarrow$|
> |:---:|:---:|:---:|:---:|:---:|
> |BasicVSR++|30|32.38|0.9070|0.1462|
> |RVRT|30|32.75|0.9113|0.1410|
> |VRT|16|32.19|0.9005|0.1544|
> |DDNM|1|27.05|0.7660|0.2608|
>
> > Q3. The authors should compare the efficiency, including model size, training/inference time and FLOPs. The efficiency comparisons can demonstrate the effectiveness of the proposed method.
>
> Please see the answer to Q2 of the global response.

---

### Author Rebuttal · Authors · 2024-08-07

Dear AC and reviewers,

We sincerely thank all reviewers for your constructive comments. We are glad that the reviewers appreciate the **novelty** (srYj, ERJ1), **writing** (ERJ1), **impressive experimental results** (vLVk, srYj, ERJ1, NXC1) and limitations adequately discussed (vLVk, srYj, ERJ1) in the paper. Since reviewer vLVk, reviewer srYj, and reviewer NXC1 all concern the ablation results for specific components and require supplementing the efficiency comparison with other related methods. Thus, we address these issues in the following parts.

> Q1. Ablation study about wavelet transform and Blurring ResShift.

We add an ablation study to validate the efficacy of the introduced wavelet transform on the REDS4 dataset. After substituting the wavelet transform with simple downsampling and upsampling, denoted as model #6 in Table 1 of the rebuttal document, a noticeable decline in the values of PSNR/SSIM can be observed compared to SeeClear. It can be drawn that wavelet transform not only changes the resolution of features like downsampling/upsampling but also assists SeeClear in achieving consistency with LR frames in the low-frequency component (indicated by a drop in PSNR value) and efficiently generating high-frequency spectrums via skip connections between the encoder and the decoder (shown by the deterioration of the perceptual metric), thereby further promoting the performance.

We also combine different degrees of blur intensity and noise schedule of ResShift to comprehensively verify the necessity of Blurring ResShift in Table 2 of the rebuttal document. It can be concluded that different options of blur intensity impact the trade-off between fidelity and perceptuality (Line 1-3), and the introduction of wavelet transform and self-attention across frequency spectrums can significantly improve both fidelity and perceptuality at the same time (Line 2 v.s. Line 5).

> Q2. Efficiency comparisons to other competing methods.

We provide a comprehensive comparison of the efficiency between our proposed method and diffusion-based methods in Table 4 of the rebuttal document. It presents the number of parameters of different models and their inference time for super-resolving 512 × 512 frames from 128 × 128 inputs. Combining these comparative results, we draw the following conclusions: i) Compared to semantic-assisted single-image super-resolution (e.g., CoSeR and SeeSR), our proposed method possesses fewer parameters and higher inference efficiency. ii) In contrast to existing diffusion-based methodologies for VSR, SeeClear is much smaller and runs faster, benefiting from the reasonable module designs and diffusion process combing patch-level blurring and residual shift mechanism.

> Q3. The rational explanation of Blurring ResShift.

Blurring ResShift, a patch-level blurring version of ResShift, is elaborated in the related works, methods, and supplementary materials. Comparisons to ResShift and ablation experiments were exhibited in the main text and supplementary materials. We also supplement additional ablation experiments for the issues that concern the reviewers. Additionally, we analyze the final states of different diffusion processes via the power spectral density, which reflects the distribution of frequency content in an image, as illustrated in Figure 2 of the rebuttal document. It can be observed that IHDM performs blurring globally and has a significant difference in frequency distribution compared to the LR image, while the patch-level blurring is closer to the frequency distribution of the LR. On this basis, SeeClear further introduces residual and noise. Compared to ResShift without blurring, the diffusion process adopted by SeeClear makes the image more consistent with the LR in the low-frequency components and introduces more randomness in the high-frequency components, compelling the model to focus on the generation of high-frequency components.

---

### Decision · Program_Chairs · 2024-09-25

**Decision:**

Accept (poster)

**Comment:**

This paper received three positive and a negative score. After checking the rebuttal, the AC agreed with three positive reviewers, considering the paper's technical contribution and performance improvement.

If the paper is accepted, please proofread the submission more to make it easier to understand. For example, the method section needs more clarification. And add more evaluation metrics such as DISTS, FID, NIQE, MANIQA, MUSIQ and CLIPIQA.